# Thermal dissociation cavity-enhanced absorption spectrometer for measuring NO₂, RO₂NO₂ and RONO₂ in the atmosphere

Chunmeng Li[1], Haichao Wang[1, 2, 3*], Xiaorui Chen[1], Tianyu Zhai[1], Shiyi Chen[1], Xin Li[1], Limin Zeng[1], Keding Lu[1, *]

[1] State Key Joint Laboratory of Environmental Simulation and Pollution Control, College of Environmental Sciences and Engineering, Peking University, Beijing, 100871, China.

[2] School of Atmospheric Sciences, Sun Yat-sen University, Zhuhai, Guangdong, 510275, China.

[3] Guangdong Provincial Observation and Research Station for Climate Environment and Air Quality Change in the Pearl River Estuary, Key Laboratory of Tropical Atmosphere-Ocean System, Ministry of Education, Southern Marine Science and Engineering Guangdong Laboratory (Zhuhai), Zhuhai, 519082, China.

* Correspondence: wanghch27@mail.sysu.edu.cn; k.lu@pku.edu.cn

**Abstract.** We developed thermal dissociation cavity-enhanced absorption spectroscopy (TD-CEAS) for the in situ measurement of NO₂, total peroxy nitrates (PNs, RO₂NO₂), and total alkyl nitrates (ANs, RONO₂) in the atmosphere. PNs and ANs were thermally converted to NO₂ at the corresponding pyrolytic temperatures and detected by CEAS at 435-455 nm. The instrument sampled sequentially from three channels at ambient temperature, 453 K and 653 K, with a cycle of 3 min, to measure NO₂, NO₂+PNs, and NO₂+PNs+ANs. The absorptions between the three channels were used to derive the mixing ratios of PNs and ANs by spectral fitting. The detection limit (LOD, 1σ) for retrieving NO₂ was 97 parts per trillion (pptv) in 6 s. The measurement uncertainty of NO₂ was 9%, while the uncertainties of PNs and ANs detection were larger than those of NO₂ due to chemical interferences that occurred in the heated channels, such as the reaction of NO (or NO₂) with the peroxy radicals produced by the thermal dissociation of organic nitrates. Based on laboratory experiments and numerical simulations, we created a look-up table method to correct these interferences in PNs and ANs channels under various ambient organic nitrates, NO, and NO₂. Finally, we present the first field deployment and compare it with other instruments during a field campaign in China. The advantages and limitations of this instrument are outlined.

# 1. Introduction

Organic nitrates (ONs) act as temporary NOx reservoir species, which affect atmospheric circulation and impact air quality and climate (Mellouki et al., 2015). Peroxy nitrates (PNs, $RO_2NO_2$) and alkyl nitrates (ANs, $RONO_2$) are two important kinds of organic nitrates. They are closely related to the distribution of oxidants in the atmosphere by terminating the HOx cycle. ONs are also important precursors of secondary organic aerosols (SOAs) (Berkemeier et al., 2016; Lee et al., 2016; Ng et al., 2017; Rollins et al., 2012). Volatile organic compounds (VOCs) are oxidized by OH or $O_3$ to produce peroxy radicals ($RO_2$), and then $RO_2$ reacts with $NO_2$ to produce PNs (R1). In addition, the aldehydes formed during the process of $NO_3$ oxidizing isoprene at night react with $NO_3$ to form PNs. PNs can be divided into two categories depending on the nature of the $RO_2$ group. One is peroxy acyl nitrates (PANs) when $RO_2$ is R'C (O)OO, among which PPN (peroxypropionyl nitrate) and PAN (peroxyacetyl nitrate) dominate PNs with percentages of 75-90% due to their relatively high thermal stability. The other is some peroxy nitrates without acyl groups, which are only abundant in cold regions (Roberts, 1990; Roberts et al., 1998b; Thieser et al., 2016; Wooldridge et al., 2010). The sink pathways of PNs include deposition, thermal decomposition, photolysis, and OH oxidation, and thermal decomposition dominates in the troposphere with a temperature dependence (R2). Therefore, the lifetime of PAN varies from less than one hour to several months, depending on the environmental conditions.

$$RO_2 + NO_2 + M \rightarrow RO_2NO_2 + M \qquad (R1)$$
$$RO_2NO_2 + M \rightarrow RO_2 + NO_2 + M \qquad (R2)$$

In the high NOx region, $RO_2$ reacts primarily with NO to produce ANs. ANs can also be emitted directly from biomass combustion and the ocean. Ocean emissions are regarded as the main source of short-chain ANs ($C_1$-$C_3$), and up to tens of pptv of the species above have been measured in marine areas (Atlas et al., 1993; Chuck et al., 2002; Talbot et al., 2000). $NO_3$-initiated ANs during the night are generally considered to be important and have a higher organic nitrate yield than OH-initiated ANs (Horowitz et al., 2007; Perring et al., 2013). During the daytime, there is a branching reaction between $RO_2$ and NO to form ANs (R3a) with a small branch ratio (1-30%) (Arey et al., 2001; Reisen et al., 2005; Russell and Allen, 2005; Wennberg et al., 2018). Ambient ANs are removed by photolysis or oxidation to produce NOx or $HNO_3$; deposition or transportation as NOx reservoirs. ANs play a significant role in SOA formation (Lee et al., 2016; Zare et al., 2018). Monofunctional ANs are stable and account for a small proportion of ANs, among which those formed from alkanes can be tracers of human activities in remote areas (Simpson et al., 2006; Wang et al., 2003). Polyfunctional ANs are hard to detect since they are more reactive than monofunctional ANs.

$$RO_2 + NO + M \rightarrow RO_2NO + M \qquad (R3a)$$
$$RO_2 + NO + M \rightarrow RO + NO_2 \qquad (R3b)$$

The various sources and sinks of ONs complicate their atmospheric distribution. The measurement of ANs and PNs has been developed by gas chromatography (GC). GC is used for the separation of species, and then the separated substances are quantified by electron capture detectors (ECD), luminol chemiluminescence (CL), or mass spectrometry (MS) (Atlas, 1988; Blanchard et al., 1993; Flocke et al., 2005; Gaffney et al., 1998; Hao et al., 1994; Luxenhofer et al., 1994; Tanimoto et al., 1999). These methods measure individual species accurately (Roberts et al., 2003), but the individual standards are incomprehensive. Furthermore, the methods suffer from relatively low time resolution (Blanchard et al., 1993). The strength of the bond between the $NO_2$ group and the organic group determines the temperature to pyrolyze the organic nitrates. The cleavage of the $NO_2$ group in PNs requires approximately 85-115 kJ/mol (Kirchner et al., 1999), while for ANs, the pyrolytic process requires approximately 160-170 kJ/mol (Roberts, 1990); therefore, PNs are more prone to dissociate thermally. Based on the feature of gradient pyrolysis of reactive nitrogen compounds, TD-LIF (thermal-dissociation laser-induced fluorescence) was developed to measure PNs, ANs, and gaseous $HNO_3$ indirectly by quantifying the $NO_2$ product (Day et al., 2002). Afterwards, chemical

ionization mass spectrometry (CIMS), cavity ring-down spectroscopy (CRDS) and cavity attenuated phase-shift spectroscopy (CAPS) are used to quantify the pyrolytic products (Paul and Osthoff, 2010; Slusher et al., 2004; Thieser et al., 2016; Wild et al., 2014). The detection limits and response times of TD-CIMS are excellent, but [13]C-labeled PAN is required as an internal standard. TD-CRDS and TD-CAPS show high spatial and temporal resolution and good measurement capability (Sadanaga et al., 2016; Sobanski et al., 2016). CEAS (cavity-enhanced absorption spectroscopy) is a powerful technology that can monitor several compounds or species simultaneously with broad absorption bands being detected (Fiedler et al., 2003) and has been applied to measure many species in field studies, such as $NO_2$, HONO, $NO_3$, $N_2O_5$, IO, glyoxal, and methylglyoxal (Ball et al., 2004; Barbero et al., 2020; Duan et al., 2018; Gherman et al., 2008; Jordan and Osthoff, 2020; Kahan et al., 2012; Langridge et al., 2006; Lechevallier et al., 2019; Liu et al., 2019; Min et al., 2016; Thalman and Volkamer, 2010; Vaughan et al., 2008; Venables et al., 2006; Ventrillard-Courtillot et al., 2010; Ventrillard et al., 2017; Wang et al., 2017a; Washenfelder et al., 2016; Washenfelder et al., 2008; Watt et al., 2009).

Organic nitrates have a large range of mixing ratios in the atmosphere that vary from several pptv in warm and remote regions to several ppbv in polluted regions. Field measurements of organic nitrates have been extensively conducted in the United States and Europe (Fischer et al., 2000; Glavas and Moschonas, 2001; Kastler and Ballschmiter, 1999; Perring et al., 2009; Roberts et al., 1998a; Sobanski et al., 2017), but related studies are sparse in China (Chen et al., 2017; Song et al., 2018; Sun et al., 2018; Zhang et al., 2018). Ozone pollution in China has occurred frequently in recent years (Ma et al., 2019; Shu et al., 2019; Wang et al., 2009; Wang et al., 2017b; Yin et al., 2019). Although many studies have examined the effect of PNs and ANs on regulating ozone formation (Chen et al., 2018; Ling et al., 2016; Liu et al., 2018; Liu et al., 2012; Liu et al., 2010; Zeng et al., 2019; Zhang et al., 2014), the issue has not been well studied. Here, we developed a pyrolytic measurement system based on CEAS to detect $NO_2$, PNs, and ANs in the atmosphere. In this study, the detailed setup of the instrument, laboratory characterizations, and its first field application in China are presented.

## 2. Methods

### 2.1 Instrumentation of TD-CEAS

Our instrument is designed to measure $NO_2$, ANs and PNs in the atmosphere, and has the characteristics of good stability, low energy consumption and portability. The total weight of the instrument is less than 30 kg, the overall size is $110 \times 60 \times 50$ cm, and power consumption is less than 300 W. The measurement of $NO_2$ is achieved by CEAS. Due to the feature of gradient pyrolysis of ANs and PNs, the sample gas flowing out from three different channels contains the total amount of $NO_2$ at different temperatures. The gradient of $NO_2$ concentration absorption at different pyrolytic temperatures is used to retrieve the mixing ratio of $NO_2$, PNs, and ANs. The time resolution of the instrument measurement is 6 s, measurement time of each channel is 1 min, and each cycle is 3 min.

The CEAS system has been described in detail in previous literature (Duan et al., 2018; Fiedler et al., 2003; Gherman et al., 2008; Jordan and Osthoff, 2020; Jordan et al., 2019; Langridge et al., 2006; Liang et al., 2019; Liu et al., 2019; Min et al., 2016; Tang et al., 2020; Ventrillard-Courtillot et al., 2010; Wang et al., 2017a; Yi et al., 2016), thus there is a brief introduction to the principle of the instrument here. $NO_2$ molecules have a specific absorption structure in the wavelength range of 430-460 nm (Fig. S1). Based on Lambert-Beer's law, the extinction coefficient ($\alpha$) is proportional to the absorber's concentration and optical path. Here, $\alpha$ is mainly contributed by molecular absorption, Rayleigh scattering and Mie scattering. In addition, it can also be obtained by comprehensive calculations through the intensity of the sampling spectrum, reference spectrum, mirror reflectivity, and effective cavity length.

In Eq. 1, $\lambda$ is the wavelength of light, $I_0(\lambda)$ is the intensity of the reference spectrum, $I(\lambda)$ is the sample spectrum, $d_{eff}$ is the effective cavity length (see Sec. 3.2 in detail), $R(\lambda)$ is the mirror reflectivity, $\alpha_{Mie}(\lambda)$ is the extinction

due to Mie scattering, $\alpha_{Rayl}(\lambda)$ is the extinction due to Rayleigh scattering, and $n_i$ and $\sigma_i(\lambda)$ are the number
density and absorption cross-section of $i$th gas compounds, respectively. According to Eq. 1, it is necessary to quantify
the mirror reflectivity, effective cavity length, and $NO_2$ absorption cross-section.
$$\alpha(\lambda) = \left(\frac{I_0(\lambda)}{I(\lambda)} - 1\right)\left(\frac{1-R(\lambda)}{d_{eff}}\right)$$
$$= \sum_i n_i \times \sigma_i(\lambda) + \alpha_{Mie}(\lambda) + \alpha_{Rayl}(\lambda) \qquad (1)$$
As shown in Fig. 1a, the optical layout of the CEAS consists of a light source, collimating optics, cage system,
high-finesse cavity and a commercial spectrograph with a charge-coupled device (CCD) detector. The core of the
light source module is a single-color LED (M450D3, Thorlabs, Newton, NJ, USA), which emits approximately
1850 mW optical power at approximately 450 nm with a full width at half maximum (FWHM) of 18 nm. To obtain
a stable output of the light source, the input current and operating temperature of the light source are stabilized to
reduce the intensity and wavelength drift. The switching power supply is 12 VDC with a current of $1.00 \pm 0.01$ A.
Constant current control is achieved through a stable current source. The temperature of the light source is controlled
by the proportion integration differentiation (PID) algorithm and stabilized at $24.0 \pm 0.1$ °C.
Four stainless steel columns are used to collimate two opposing high-mirror mounting bases. The two endplates in
the middle of the cage structure further enhance the stability of the system. The light source is introduced into the
system through a fiber connected to a two-dimensional adjustment frame (CXY1, Thorlabs, Newton, NJ, USA)
through a connector. The plano-convex lens (f = 30 mm) is installed in another adjustment frame, and the two
adjustment frames are connected by a customized X-shaped adapter, which is fixed at the end with the light source.
The center alignment of the light source, lens, and high-reflectivity module is achieved by adjusting the adjustment
frame in the vertical and horizontal directions. Then, blue light is introduced into an optical cavity composed of a
pair of high-reflectivity (HR) mirrors. The reflectivity of HR mirrors (CRD450-1025-100, Advanced thin films, CO,
USA) is reported by the manufacturer to be greater than 0.9999 (440-460 nm) with a radius curvature of 1.0 m and a
diameter of 25.4 mm. The high-reflectivity mirrors are installed in the groove of the special customized base and
sealed by an O-ring, and then the three-dimensional microadjustment is achieved by squeezing the lens and O-ring
to finely adjust their pitch and yaw. The distance between mirrors is 39.0 cm and high-purity nitrogen (> 99.999%),
which passes through the small hole before the mirror base, is used as a purge gas to protect the mirror surface.

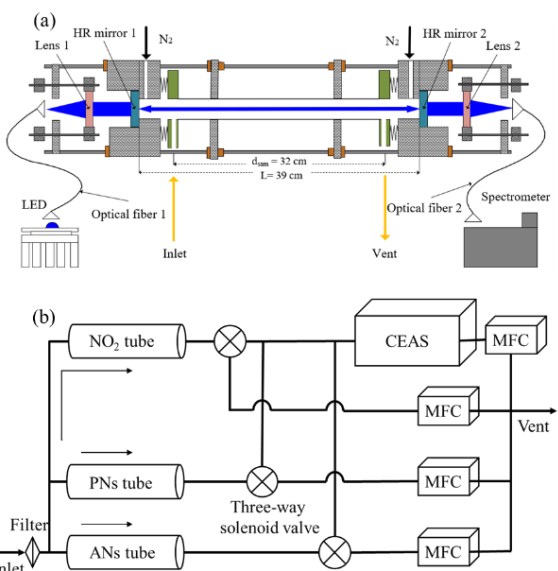


Figure 1. The overall schematic of the CEAS (a) and the instrument (b). The CEAS is mainly composed of LEDs, collimating optics, a cage structure, a high-finesse cavity and a spectrometer. After filtering the PM, the gas passes through three quartz tubes, and then the alternate measurements of $NO_2$, $NO_2$+PNs and $NO_2$+PNs+ANs are achieved by three-way solenoid valves.

The cavity system is sealed by two welded bellows, two polytetrafluoroethylenes (PTFE) connecting pieces, and a stainless-steel sampling cell that is internally polished. The PTFE connecting piece connects the sampling cell and bellows and acts as a sample inlet and outlet. As shown in Fig. 1(a), the distance between the inlet and outlet ($d_{sam}$) is 32.0 cm. After passing through the sampling cavity, the blue light converges through another plano-convex mirror ($f$ = 50 mm). It enters the detector spectrometer (QE65PRO, Ocean Optics, Dunedin, FL, USA) for signal acquisition through an optical fiber. The dark current in the CCD of the spectrometer is reduced by controlling the temperature of the CCD at -20.0 °C; the width of the entrance slit is 100 μm, the corresponding wavelength resolution is 0.39 nm, and the detection wavelength range is 413.48-485.48 nm.

The schematic of the TD-CEAS is shown in Fig. 1(b). The flow system mainly includes the particulate matter filter in the front end of the sampling line, quartz tubes for species pyrolytic conversion, three-way system switching module, detection module (CEAS) and flow control module. The sample gas first passes through a PTFE filter membrane (25 μm thickness, 4.6 cm diameter, and 2.5 μm pore size, Typris, China) to remove ambient aerosols. The sample gas enters the system through a 1/4' inch PFA (polytetrafluoroethylene) tube and is then divided into three channels ($NO_2$ channel, ANs channel, and PNs channel) by using two T-shaped PFA three-way connections. The gas flow at the end of each channel is controlled at 0.8 L/min, and the total flow rate (sample flow gas and purge gas) is 2.6 L/min maintained by mass flow controllers and a diaphragm pump.

The quartz tubes have a length of 35 cm, which have an inner diameter of 5 mm and an outer diameter of 10 mm and are connected to the system through a 10 mm to 1/4' inch PTFE connection. The quartz tubes of the ANs channel and the PNs channel are heated by resistance wires, and temperature is controlled by the PID algorithm. An asbestos sleeve on the quartz tube surface is used to insulate heat exchange with the external environment. The heating powers of the PNs channel and ANs channel are approximately 20 W and 50 W, respectively. The length of the heating module is 15 cm. According to the pyrolytic efficiency experiment (see Sec. 3.4 for details), the heating temperatures for the ANs and PNs channels are controlled at 380 °C and 180 °C, respectively. One CEAS is used to detect the $NO_2$ absorption of different channels to reduce the cross-interference due to the difference of multidetectors. A solenoid valve is connected behind the quartz tube of each channel. At the same time, a time relay is used to periodically control the three T-shaped solenoid valves (71335SN2KVJ1, Parker Hannfin, USA), and the internal surface of the T-shaped solenoid valves is stainless steel. Each channel has the same constant flow rate regardless of whether the sampling air draws into the CEAS or vent. At the end of the channels, mass flow controllers are used to restrict the flow rate.

## 2.2 Laboratory experimental setup

To characterize the performance and potential interferences of this instrument, we used a photochemical PAN source in the laboratory experiments. Acetone undergoes photolysis at 285 nm from a Hg lamp and then generates excess PA radicals (peroxyacetyl radicals) in zero air. A small amount of NO reacts with PAs to form $NO_2$, and then $NO_2$ further reacts with PAs to form PAN. We obtained a standard PAN source in this way, which generated a source at a level of 1-10 ppbv. The source was used for the laboratory experiments after the temperature of the Hg lamp stabilized at 39.0 °C, and the source level and stability were double-checked by a GC-ECD instrument. To investigate the potential interferences caused by the pyrolysis of organic radical products reacting with ambient NO and $NO_2$ in the TD-CEAS, a multigas calibrator (146i, Thermo Fisher Scientific, Inc., USA) was used to generate $O_3$ gas by photolysis of oxygen and outputted well-mixed gases by diluting NO or $NO_2$ with zero air according to the

requirement of studying the potential inferences caused by ambient NO and $NO_2$. NO (1 ppmv) and $NO_2$ (10 ppmv)
bottle gases were connected to the multigas calibrator. An ozone monitor was used to detect $O_3$ levels in these
experiments (49i, Thermo Fisher Scientific, Inc., USA). A $NO_x$ monitor was used to detect NO and $NO_2$ levels in
these experiments (42i, Thermo Fisher Scientific, Inc., USA). Pure $N_2$ (>99.9999%) and He (>99.9999%) bottle gases
were used to calibrate the mirror reflectivity of the CEAS and to purge the mirrors.

## 2.3 Box model

A box model was established to mimic the experimental results and study the potential interferences of NO and $NO_2$
in the PNs and ANs measurements. The chemical mechanism is based on previous work (Thieser et al., 2016). These
reactions during the pyrolytic process in the box model are listed in Text S1, and the reaction rate of these reactions
is mainly taken from the Master Chemical Mechanism, MCM v3.3 (website: http://mcm.leeds.ac.uk/MCM) (Jenkin
et al., 1997; Saunders et al., 2003). As the wall loss has an important effect on the lifetime of free radicals, we set the
wall loss constant ($k_{wall}$) of $RO_2$ to 0.3/s (Thieser et al., 2016; Wooldridge et al., 2010). The wall loss rate coefficients
of $HO_2$ and OH are selected as the values of 0.5 and 5.4/s, respectively (Fuchs et al., 2008). The residence time of
the sampling gas in each channel is calculated by considering the temperature distribution. The time step of the model
is set to 0.001 s.

## 3. Instrument characterization

### 3.1 Mirror reflectivity

The spectra of pure $N_2$ (>0.99999) or He (>0.99999) filling the cavity through the purge lines are collected to calibrate
the mirror reflectivity, as the Rayleigh scattering section of the two is significantly distinct; therefore, $R(\lambda)$ can be
calibrated according to Eq. 2 (Chen and Venables, 2011; Min et al., 2016).

$$R(\lambda) = 1 - d \times \left( \frac{I_{N_2}(\lambda) \times n_{N_2} \times \sigma_{Rayl,N_2}(\lambda) - I_{He}(\lambda) \times n_{He} \times \sigma_{Rayl,He}(\lambda)}{I_{He}(\lambda) - I_{N_2}(\lambda)} \right) \qquad (2)$$

where $d$ is the distance between two high-reflectivity mirrors, $\lambda$ is the wavelength, $I_{N_2}(\lambda)$ and $I_{He}(\lambda)$ are spectra
obtained when the cavity is filled with pure $N_2$ and He, respectively, $n_{N_2}$ and $n_{He}$ are the number densities calculated
at the measurement temperature and pressure in the cavity, respectively, and $\sigma_{Rayl,N_2}(\lambda)$ and $\sigma_{Rayl,He}(\lambda)$ are the
Rayleigh scattering sections of $N_2$ and He, respectively (Shardanand, 1977; Sneep and Ubachs, 2005). Fig. 2 shows
the average of the mirror reflectivity calibration results. $R(\lambda)$ is above 0.9999 at 435-465 nm and up to 0.99992 at 450
nm. The total uncertainty of the mirror reflectivity is 5%, which comes from the uncertainty in the scattering section
of $N_2$. The blue line is the average optical path length when the sampling flow rate in the cavity is 0.8 L/min, which
is equal to $d_{eff}/(1-R)$ ($d_{eff}$ is 31.84 cm), with a value up to 5.2 km at 450 nm.

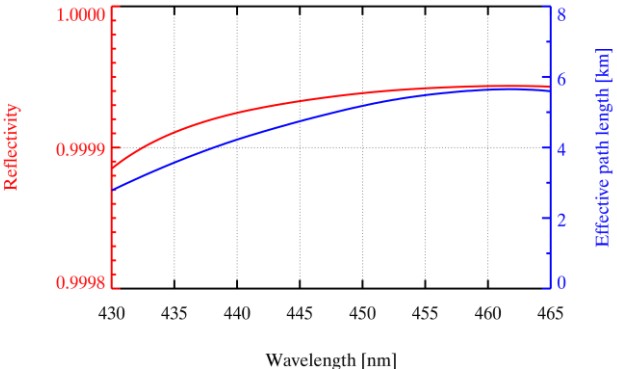


Figure 2. Mirror reflectivity and optical path length calibrated by He (>0.99999) and N₂ (>0.99999). The red line is the average $R(\lambda)$ and the blue line is the optical path length.

## 3.2 Effective cavity length

The effective length of the absorbers (named effective cavity length, $d_{eff}$) in the detection cell is shorter than the physical distance of the cavity with purge; thus, it needs to be calibrated. We performed concentration determination on the $NO_2$ standard source (130 ppbv) under two experimental settings with or without purging and then used Eq. 1 to calculate $d_{eff}$. The ratio of $NO_2$ absorption with and without purging is equal to the ratio of the effective cavity length to the physical distance $L$ between the mirrors ($d_{eff}/L$). A $NO_2$ stand was prepared from a bottled standard (5 ppmv $NO_2$) and diluted with high-purity $N_2$ in a multigas calibrator (146i, Thermo Fisher Scientific, Inc., Waltham, MA, USA). As shown in Fig. 3a, the retrieved concentration of $NO_2$ shows a general positive correlation trend with the flow rate with $N_2$ purging; the concentration of $NO_2$ is 130 ppbv without a purge. The $d_{eff}/L$ at different sampling flow rates is shown in Fig. 3b. The ratio of the effective cavity length increases as the flow rate increases, suggesting the importance of airflow rate stability during sampling. The uncertainty of the prepared $NO_2$ standard source is estimated to be 2.0%, while the uncertainty of the $NO_2$ absorption cross-section is 4.0%, according to Voigt et al. (2002). As a result, the total uncertainty of $d_{eff}$ calibration is 4.5% (Voigt et al., 2002).

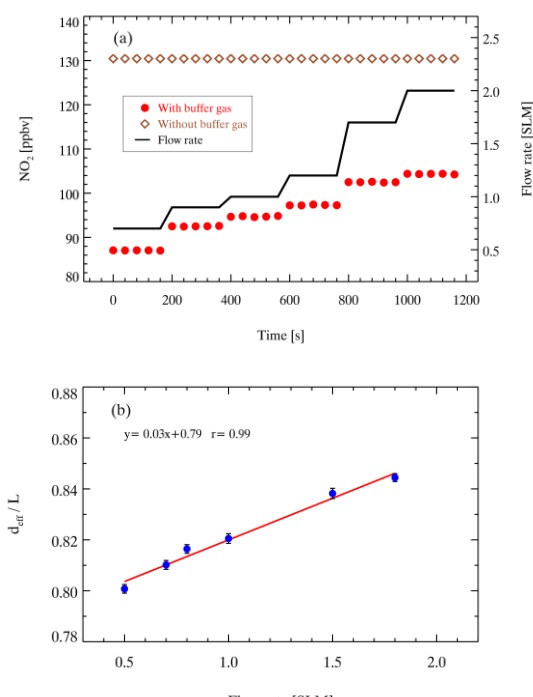

Figure 3. The results of the effective cavity length. Panel (a) The black line represents the flow rate and the red points and brown diamonds represent the retrieved $NO_2$ concentration with and without nitrogen purge (100 sccm×2), respectively. Panel (b) The relationship between the ratio of the effective cavity length ($d_{eff}$) to cavity physical distance ($L$) and the sampling flow rate.

## 3.3 Spectral fitting

The absorption cross-section of $NO_2$ measured by Voigt et al. (2002) is used to retrieve the $NO_2$ concentration in this study. The absorption cross-section of $NO_2$ between 435-455 nm is selected to perform the spectral fitting. It has been reported that the $NO_2$ cross-section is not sensitive to temperature changes (Vandaele et al., 2002; Voigt et al., 2002); therefore, convolution is only performed for our instrument setup at ambient temperature. The peak at 436.2 nm of

the Hg spectrum measured by the spectrometer is used to generate a wavelength-dependent instrument slit function that accounts for the change in spectral resolution over the CCD pixels. The convoluted cross-section of $NO_2$ is shown in Fig. S1. The measured absorption coefficient ($\alpha$) is processed by the DOASIS (Differential Optical Absorption Spectroscopy Intelligent System). The fitting shift is constrained within $\pm0.2$ nm. Glyoxal has strong absorption in the same optical window (Liu et al., 2019; Min et al., 2016; Thalman et al., 2015; Thalman and Volkamer, 2010; Washenfelder et al., 2008), but here, we do not take glyoxal absorption into consideration in the spectral fitting. The inclusion of glyoxal in the spectral fitting would enlarge the fitting residual. Our field measurements showed that the uncertainty caused by excluding glyoxal fitting was approximately 4% (Fig. S2). Fig. 4 shows two examples of the spectral fitting of the measured absorption of high and low $NO_2$ at a 6 s integration time during the ambient measurement. The retrieved mixing ratios of $NO_2$ were $16.2 \pm 0.1$ ppbv and $1.8 \pm 0.1$ ppbv, respectively. The corresponding fitting residual, which is the difference between the measured and fitting results, is in the range of $10 \times 10^{-9}$ at 435-455 nm. A typical measurement sequence during the ambient measurement is illustrated in Fig. 5, which displays $NO_2$ mixing ratios of three channels alternatively. The mixing ratio of $NO_2$ in different channels is detected periodically, and there are several transitional points due to switching measurement phases. Therefore, we excluded the transition point of each phase and the two data points before and after the transition point to avoid measurement error. As we discuss later, the mixing ratio of ANs and PNs can be calculated by subtracting the $NO_2$ mixing ratio measured from different channels.

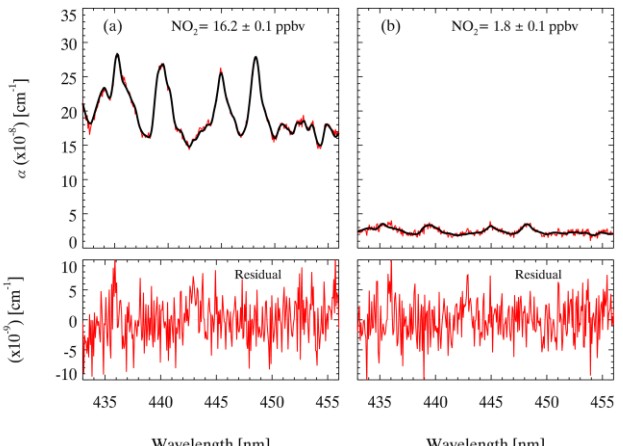

Figure 4. An example of the spectral fit for an extinction spectrum measured (6 s average) during field measurements. The fitted results of $NO_2$ are shown, and the total fit result and the residual at high concentrations (a) and low concentrations (b) are shown.

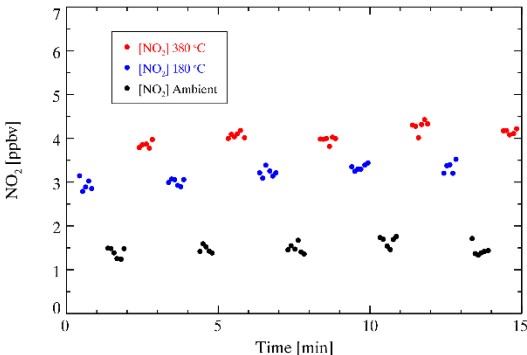

Figure 5. An example of typical measurements performed in a field study with a 6 s spectrum integral time. A measurement cycle includes
three phases whose duration is 60 s. The red points denote the $NO_2$ mixing ratio measured in the ANs channel ([$NO_2$] 380°C), the blue
points denote the $NO_2$ mixing ratio measured in the PNs channel ([$NO_2$] 180°C), and the black points denote the $NO_2$ mixing ratio
measured in the reference channel ([$NO_2$] ambient temperature).
There are two methods to determine the mixing ratio of ONs and PNs. One is the differential concentration method
('CONC'). As shown in Eqs. 3-6, the $I_0$ is fixed during data analysis by using the $N_2$ spectrum: $I_{TD380}$ and $I_{TD180}$ are
the spectra obtained when the CEAS detects the ANs channel and PNs channel, respectively; $I_{N_2}$ is the $N_2$ spectrum
obtained when the cavity is filled with $N_2$ (>0.99999); $\alpha_{TD380}$ and $\alpha_{TD180}$ are absorption coefficients when setting
$I_{N_2}$ as $I_0$, and setting $I_{TD380}$ or $I_{TD180}$ as I, respectively; and after deleting the abnormal points caused by phase
switching, [ONs] is obtained by subtracting [$NO_2$]$_{TD380}$ from the average of [$NO_2$]$_{REF}$, and [PNs] is obtained by
subtracting [$NO_2$]$_{TD180}$ from the average of [$NO_2$]$_{REF}$. The other method is the differential absorption method ('SPEC'),
by using the dynamic background spectrum method for spectral fitting (Eqs. 7-8): $I_{REF}$ is the spectrum obtained at
the reference channel; ONs can be retrieved based on $I_{TD380}$ and $I_{REF}$; and PNs can be retrieved by $I_{TD180}$ and $I_{REF}$.
An intercomparison of field measurements shows that the 'SPEC' method results in fewer outliers (Fig. 6). For the
'SPEC' method, the shift and squeeze of the spectrum is performed only once during the spectral fitting, which reduces
the uncertainty caused by the second spectral fitting. Therefore, we selected the 'SPEC' method to retrieve the
concentrations of $NO_2$, PNs, and ANs in the following data processes.
$$\alpha_{TD380} = \left(\frac{I_{TD380}}{I_{N_2}}-1\right)\left(\frac{1-R(\lambda)}{d_{eff}}\right) \qquad (3)$$
$$\alpha_{TD180} = \left(\frac{I_{TD180}}{I_{N_2}}-1\right)\left(\frac{1-R(\lambda)}{d_{eff}}\right) \qquad (4)$$
$$[ONs] = [NO_2]_{TD380}-[NO_2]_{REF} \qquad (5)$$
$$[PNs] = [NO_2]_{TD180}-[NO_2]_{REF} \qquad (6)$$
$$\alpha_{[ONs]} = \left(\frac{I_{TD380}}{I_{REF}}-1\right)\left(\frac{1-R(\lambda)}{d_{eff}}\right) \qquad (7)$$
$$\alpha_{[PNs]} = \left(\frac{I_{TD180}}{I_{REF}}-1\right)\left(\frac{1-R(\lambda)}{d_{eff}}\right) \qquad (8)$$

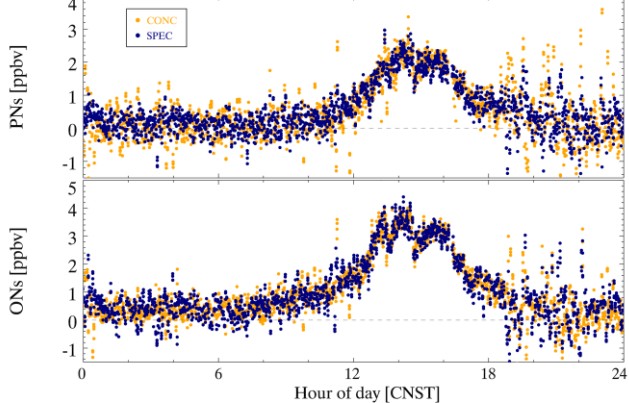


Figure 6. An example of the calculation results of the fixed $I_0$ ('CONC') and dynamic $I_0$ ('SPEC') methods performed in the field
measurements. Orange points represent the results of the 'CONC' method, and dark blue points represent the calculation results of the
'SPEC' method.
**3.4 The efficiency of thermal dissociation**

For the pyrolytic measurement of organic nitrates, the exact temperature setting for complete pyrolysis varies, mainly due to the many factors that affect the efficiency of thermal dissociation, such as the specificity of the quartz tube, the heating residence time, and the temperature distribution of the heating part (Womack et al., 2017). The thermal dissociation of PAN was tested separately in the PNs channel and ANs channel, and the efficiency curves were the same. The heating temperature is the temperature of the quartz tube surface rather than the airflow temperature in the quartz tube. The experiments were performed under normal sampling conditions, and the heating temperature was evaluated from room temperature to 440 °C to determine the appropriate heating temperature. Fig. 7 shows that the pyrolysis of PAN starts when the heating temperature is approximately 50 °C. The curve seems to reach a plateau when the heating temperature is approximately 180 °C. However, the normalized signal of thermal dissociation of PAN reaches the final plateau once the temperature is above 360 °C. Similarly, PAN is reported to be thermally dissociated completely at approximately 400 °C (Friedrich et al., 2020). The presence of alkyl nitrates in the PAN source has been reported before by previous studies and was regarded as the reason for the dual-plateau profile of PNs dissociation (Paul et al., 2009). Here, we cannot rule out the possibility of alkyl nitrate impurities. However, the source level of PAN is equal to $92 \pm 3\%$ of NOx input, suggesting only a very small percentage ($\leq 8\%$ on average), if any, of ANs. If the PAN source is equal to 4 ppbv in the PNs channel at 180 °C, as Fig. S3 shows, PAN will first dissociate completely, and then PAs will recombine with $NO_2$ to form PAN when the air flow passes through the cooling lines.

To further study the thermal dissociation of organic nitrates in the heated channels, box model simulations were conducted to reproduce the response relationship between heating temperature and $NO_2$ generated by pyrolysis. If the PAN source is equal to 4 ppbv in the PNs channel at 180 °C, as Fig. S3 shows, PAN will first dissociate completely, and then PAs will recombine with $NO_2$ to form PAN when the air flow passes through the cooling lines. As shown in Fig. 7, PAN gradually transforms into $NO_2$ and $CH_3O_2NO_2$ as the setting temperature increases. The simulated signals of thermal dissociation of PAN show two plateaus, which is generally consistent with the experimental results. However, there are some differences from 260 °C to 360 °C, which may come from simulation uncertainties such as the temperature profile in the heated channel, the follow-up reactions of PA radicals and their reaction rates. The first plateau at 180 °C is caused by the recombination of PAs and $NO_2$ after the pyrolysis of PAN, and the time for recombination from the end of the tube to the inlet is 297 ms. Therefore, recombination cannot be ignored when the heating temper plateau period indicates that almost all PAN is transformed into $NO_2$, which is due to the increase in the pyrolytic loss of PAs, and the pyrolysis of PAN is enhanced with increasing temperature.

The occurrence of the dual-plateau phenomenon is due to the competition of pyrolysis and recombination reactions. PAN will produce $NO_2$ and PAs after thermal dissociation, but PAs will recombine with $NO_2$ if PAs are not lost on the wall surface in time (R7-R8). Therefore, the thermal curve shows two plateaus as the heating temperature increases. The concentration of PAN source and wall loss rate of $RO_2$ influence the fraction of related species. Fig. 8(a) and (c) show that the gap between the two plateaus increases with PAN concentration and decreases as the wall loss rate coefficients of $RO_2$ increase. The wall loss of $RO_2$ competes with the recombination of PA radicals and $NO_2$. Therefore, a high wall loss rate coefficient of $RO_2$ reduces the recombination for PAN.

The consistency between the observed and simulated thermal efficiencies of PAN suggests that the model simulation is reliable. Without the ANs source to quantify the thermal efficiency of ANs, we try to use the model simulation to determine the heating temperature of the ANs channel. Based on the same parameter settings of the model, MeN (methyl nitrate, $CH_3NO_3$) is selected as the representative ANs to simulate the pyrolytic efficiency curve. Fig. 8(b) and (d) show that MeN can be totally thermally dissociated when the temperature is over 380 °C, indicating that a temperature set to 380 °C for the ANs channel is reasonable. The simulation also showed that the two factors have almost no effect on the pyrolysis of MeN, which is completely pyrolyzed to produce $NO_2$ when the temperature is 380 °C. The simulation results agree with previous reports about the temperature setting of thermal dissociation of

ANs, with a range from 350 °C to 450 °C (Day et al., 2002; Sadanaga et al., 2016; Sobanski et al., 2016; Thieser et al., 2016). Therefore, 180 °C and 380 °C are selected as the heating temperatures of the PNs channel and ANs channel, respectively. In addition, the interference of incomplete dissociation for PAN in the PNs channel at 180 °C is considered in the look-up table for correction, which is detailed in Sect. 4.1.

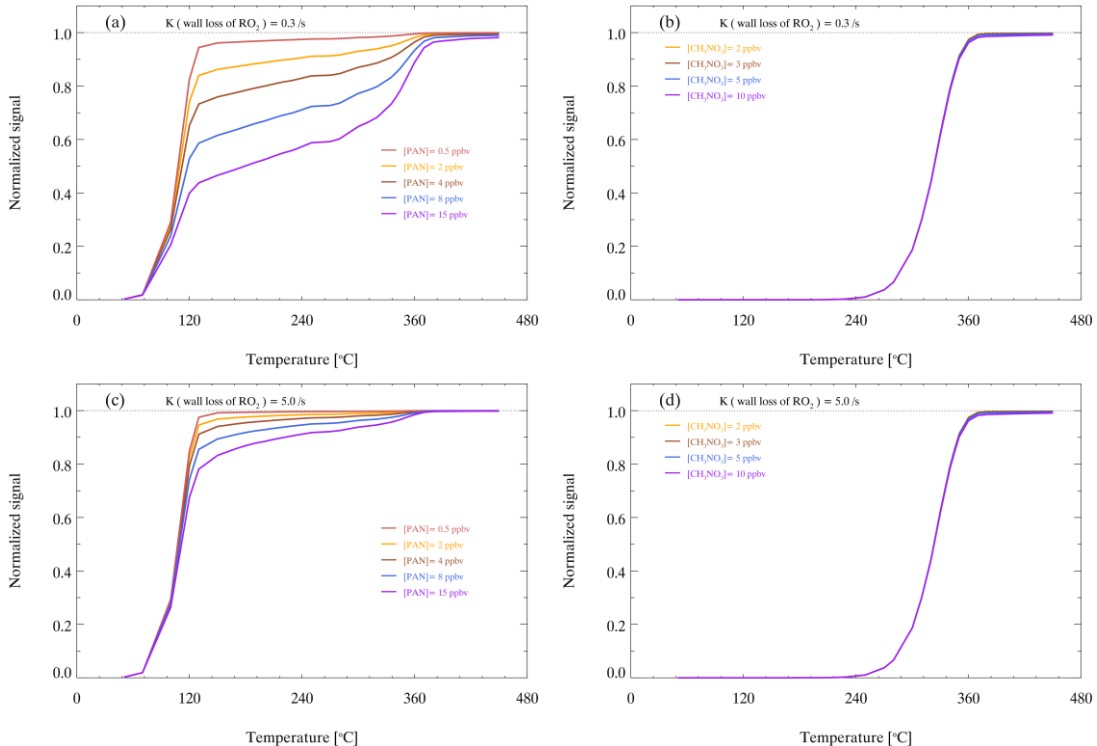

Figure 7. Normalized signals of thermal dissociation of PAN. The blue points represent the normalized signal of the observed NO$_2$ mixing ratio during thermal dissociation. The histogram represents the simulated distribution of thermal dissociation products at different temperatures, in which the gray, green and orange columns represent PAN, NO$_2$ and CH$_3$O$_2$NO$_2$, respectively.

Figure 8. Model-simulated thermal decomposition profiles of PAN and MeN with different amounts of PAN or MeN under different wall loss rate coefficients of RO$_2$. Panels (a) and (b) show the NO$_2$ signals of PAN and MeN when the wall loss rate coefficient of RO$_2$ is 0.3/s. Panels (c) and (d) show the NO$_2$ signals of PAN and MeN when the wall loss rate coefficient of RO$_2$ is 5/s.

## 4. Results and discussion

## 4.1 Measurement interference

Previous studies have shown that the filter losses and wall losses of $NO_2$, PNs and ANs are small when using Teflon tubes and Teflon filters (Paul et al., 2009; Thieser et al., 2016). As shown in Fig. S4, the response to concentration changes of PAN was nearly instantaneous under normal sampling, suggesting that the memory effects on the inlet and cavity tubing were insignificant. As shown in Figs. S5 and S6, the filter loss and sampling tube are negligible. The transmission efficiency for PAN is > 97% if there is a fresh filter membrane in the holder. We propose that changing the filter once a day can ensure a high transmission efficiency of the species to be detected. However, isoprene nitrates are prone to hydrolysis (Vasquez et al., 2020), which is more likely to be lost during sampling. We have no evaluation of the sampling loss of isoprene nitrates, and the wall loss of isoprene nitrates is likely to be reduced by increasing the frequency of filter changes. In the heated channels, organic nitrates will be thermally dissociated to produce $NO_2$, but some simultaneous reactions will affect the $NO_2$ mixing ratio. The potential interferences mainly come from the following reactions: formation of $NO_2$ via NO and $O_3$, pyrolysis of $O_3$, reactions of organic radicals with NO and $NO_2$, and pyrolysis of other reactive nitrogen oxides.

The formation of $NO_2$ in a dark reaction between NO and $O_3$ should be considered in $NO_2$ measurements. If the reaction has continued for a certain time ($t$) during sampling, the amount of $NO_2$ formed $[NO_2]t$ can be calculated: $[NO_2]t = k \times [NO] \times [O_3] \times t$, where $k$ is the rate coefficient for reaction (R4) and is given as $2.07 \times 10^{-12} \exp(-1400/T)$ $cm^3$/molecule/s (Atkinson et al., 2004). According to the temperature distribution and airflow temperature measurements changing with the distance after the heating quartz tube, the heated channel temperature profiles under normal sampling are shown in Fig. S7. Based on the temperature profile, the reaction of NO and $O_3$ in the three channels can be calculated. As the residence time of airflow in the three channels is short and similar (0.806 s in the reference channel; 0.697 s in the ANs channel; and 0.730 s in the PNs channel), the simulation results show that the inference is small. For example, during an ozone pollution day with $O_3$ = 100 ppbv, NO = 2 ppbv and $NO_2$ = 5 ppbv, the $NO_2$ produced by the reaction of NO and $O_3$ in the reference channel is 0.07 ppbv, corresponding to 1.3% of atmospheric $NO_2$. Similarly, the interferences in the ANs channel and PNs channel are 0.14 ppbv (2.7% of $NO_2$) and 0.10 ppbv (2.0% of $NO_2$), respectively. The interferences are within 3% in the typical case, which is smaller than the uncertainty of the $NO_2$ measurement. Therefore, the interference is ignored in the measurement correction.

The thermal degradation of $O_3$ occurs at high temperatures, which reduces $NO_2$ to NO via $O(^3P)$ (R5-R7). Interference has been ignored before in the process of PNs and ANs pyrolysis (Day et al., 2002). However, subsequent studies have shown that the reaction can cause significant negative deviations in the measurements of $NO_2$ at higher temperatures, and the degree of interference is closely related to the temperature change of the pyrolytic module (Lee et al., 2014; Thieser et al., 2016). To determine the reduction reaction effect, we performed experiments in which $NO_2$ was detected in three different channels when various amounts of $NO_2$ and $O_3$ were added. The experimental results are shown in Table 1 with various amounts of $NO_2$ and $O_3$ added. No significant $NO_2$ mixing ratio difference was observed between the reference and ANs channels. We showed negligible interference here, which is different from previous reports (Lee et al., 2014; Thieser et al., 2016). This is likely caused by the much lower temperature setting of our ANs measurement channel. Since the pyrolytic rate constant of $O_3$ is highly temperature-dependent, the lower temperature would largely reduce the level of O atoms as well as this interference.

$$NO + O_3 \rightarrow NO_2 \qquad\qquad\qquad (R4)$$

$$O_3 \rightarrow O + O_2 \qquad\qquad\qquad (R5)$$

$$O + O_2 + M \rightarrow O_3 + M \qquad\qquad\qquad (R6)$$

$$O + NO_2 \rightarrow O_2 + NO \qquad\qquad\qquad (R7)$$


Table 1. Measurements of the $NO_2$ mixing ratio in three channels of the TD-CEAS with different added amounts of $NO_2$ and $O_3$.

| Order | $[NO_2]$ [ppbv] | $[O_3]$ [ppbv] | $[O_3]\times[NO_2]$ [ppbv×ppbv] | $[NO_2]_{TD380}$ [ppbv] | $[NO_2]$ -$[NO_2]_{TD380}$ [ppbv] |
|---|---|---|---|---|---|
| 0 | 7.45±0.27 | 48.19 | 359 | 7.79±0.27 | -0.34 |
| 1 | 7.89±0.27 | 67.47 | 532 | 8.17±0.28 | -0.28 |
| 2 | 15.58±0.29 | 48.19 | 751 | 15.84±0.28 | -0.26 |
| 3 | 8.23±0.27 | 96.38 | 793 | 8.22±0.28 | 0.01 |
| 4 | 15.77±0.25 | 67.47 | 1064 | 15.94±0.27 | -0.17 |
| 5 | 8.43±0.27 | 144.57 | 1218 | 8.64±0.28 | -0.21 |
| 6 | 16.18±0.28 | 96.38 | 1559 | 16.20±0.28 | -0.02 |
| 7 | 16.28±0.30 | 144.57 | 2354 | 16.26±0.31 | 0.02 |


The $RO_2$ recombines with $NO_2$ or reacts with NO to interfere with the measurement of ANs and PNs. Taking PAN
as an example, organic radicals may trigger interference, as described below (R8-R15). The PAs produced after
thermal dissociation of PAN (R8) can recombine with $NO_2$ (R9). PAs can oxidize NO to produce $NO_2$ while
generating another organic radical (R10). $CH_3O_2$ can further initiate a series of reactions that affect the distribution
of $NO_2$ (R11-R15). Therefore, the lifetime and fate of PAs generated by PAN pyrolysis will cause interference, and
atmospheric NO and $NO_2$ will affect the degree of measurement interference.
$CH_3C(O)O_2NO_2 + M \rightarrow CH_3C(O)O_2 + NO_2 + M$ (R8)
$CH_3C(O)O_2 + NO_2 + M \rightarrow CH_3C(O)O_2NO_2 + M$ (R9)
$CH_3C(O)O_2 + NO (+ O_2) \rightarrow NO_2 + CH_3O_2 + CO_2$ (R10)
$CH_3O_2 + NO \rightarrow 0.999 \times (CH_3O + NO_2) + 0.001 \times CH_3O_2NO_2$ (R11)
$CH_3O \rightarrow HCHO + HO_2$ (R12)
$CH_3O_2 + NO_2 \rightarrow CH_3O_2NO_2$ (R13)
$HO_2 + NO \rightarrow OH + NO_2$ (R14)
$OH + NO_2 \rightarrow HNO_3$ (R15)
$CH_3NO_3 \rightarrow CH_3O + NO_2$ (R16)
A set of laboratory experiments was conducted to measure the interference in PNs channels with different NO,
$NO_2$ and PAN levels. Fig. 9(a) shows the measured and simulated results of different PAN concentrations mixed
with different concentrations of NO. With the increase in NO added, the detected PN mixing ratio (the signal
difference between the PNs channel and REF channel) also gradually increased. More NO reacted with PA radicals
generated during thermal dissociation to produce additional $NO_2$ within the residence time in the PNs channel, which
led to measurements higher than the PAN source level. In contrast, as shown in Fig. 9(b), the measured PNs were
lower than the PAN source level when $NO_2$ was added to the source, and the bias increased with the increase in $NO_2$
added. The addition of $NO_2$ to the system improves the overall concentration of $NO_2$ in the PNs channel, promoting
regeneration to PAN.
We conducted numerical simulations by a box model to mimic observations and to check the chemical reactions
in the PN channel. Fig. 9(a) shows that the simulations have good consistency with the experimental results under
different NO levels. Fig. 9(b) shows that the model can capture the trend of experimental results on $NO_2$ interferences,
except in the case of a PAN source of 4.5 ppbv, which may be due to the reaction of small excess $RO_2$ in the PAN
source line and extra $NO_2$ added to the instrument to generate additional PAN before sampling. Overall, these
experiments proved that NO and $NO_2$ interfere with the measurement of PNs. However, the agreement of the
experimental and model results indicates that the interference of NO and $NO_2$ for PNs measurements can be corrected.

In the field measurements, the correction factor refers to the ratio between the real value and the measured value of PNs. For example, in a typical case during field measurements where PNs = 3 ppbv, NO = 8 ppbv and $NO_2$ = 5 ppbv, the difference between the PNs channel and reference channel is equal to 4.54 ppbv (equivalent to the measured PNs), which requires a correction factor of 0.66. Here, nearly 40000 simulations are performed under various initial concentrations of NO (0-70 ppbv), $NO_2$ (0-60 ppbv) and PAN (0-10 ppbv) to obtain the correction factor look-up table for our first field measurement (detailed in Sect. 4.3). The correction factor ($C1$) for PNs measurements in the PNs channel can be determined from the look-up table according to atmospheric NO and $NO_2$ and the raw data of PNs measurements using linear interpolation. According to Eq. 9, the corrected PNs mixing ratios are derived by the raw PNs measurements ($[NO_2\_180]$) and $C1$.

$$[PNs\_real] = [NO_2\_180] \times C1 \qquad (9)$$

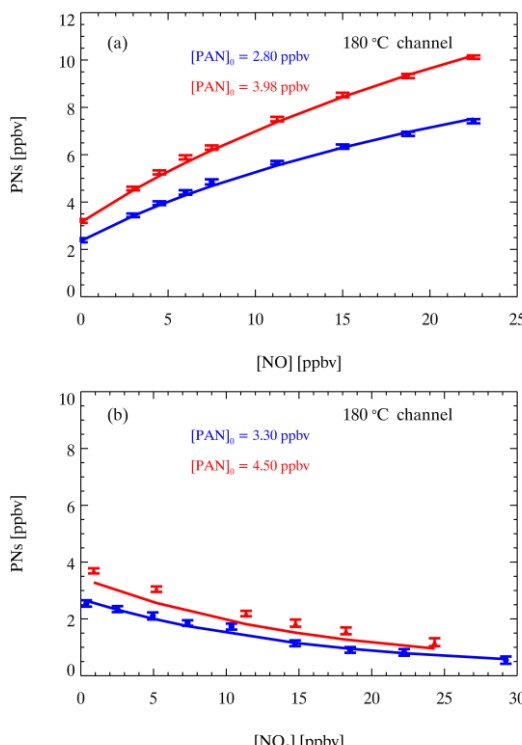

Figure 9. Simulated (lines) and measured (points) differences between the $NO_2$ signal in the PNs channel and reference channel for different PAN samples with different added amounts of NO (a) and $NO_2$ (b). The error bars show one standard deviation.

In the ANs channel, the thermal dissociation of PA radicals is rapid, and the regeneration of PAN is also suppressed at higher temperatures. Therefore, there is a different relationship between PANs and ambient NOx in the ANs channel. Similar to PNs, the measurement of ANs could be affected by NOx. We derived the PNs corrected by $C1$ and $[NO_2\_180]$ as mentioned above. To determine the corrected concentrations of ANs as Eqs. 10-11 show, we need to apply another correction factor ($C2$) to determine the contribution of PNs to ONs, in which this is subtracted from the raw ONs measurement ($[NO_2\_380]$) and finally the third correction factor ($C3$) is applied to obtain corrected ANs measurements. Fig. 10 displays the interference of NO and $NO_2$ with the PAN measurements in the ANs channel. The laboratory experiments showed that the measured signal difference increased with NO (Fig. 10(a)). Hence, the presence of NO still led to higher measurement results of ONs compared with the source value. However, the interference was weakened compared with the measured results in the PNs channel at the same NO and PAN source levels (Fig. 9(a)). Similarly, the experiments with added $NO_2$ showed underestimated measurements of ONs, and the

interference was significantly weakened compared with that in the PNs channel. We used the same box model except for updating the temperature distribution and the corresponding residence time in the ANs channel to simulate the interference of NO or $NO_2$ under different PAN source levels. However, there are still some uncertainties about the reaction mechanism and reaction rate for the thermal dissociation of PAN at these high temperatures. We performed sensitivity tests on the follow-up reactions of PAs, similar to the simulation results of Thieser et al. (2016), and found that the isomerization of PAs to $CH_2C(O)OOH$ has a great effect on the consistency of the experiments and simulation results. If the reaction rate of the branching reaction is set to zero, as shown in Fig. 10, the simulation results capture the trend well when $NO_2$ or NO is added. Many factors affect the $NO_2$ signal produced by the thermal dissociation of PAN; therefore, the mechanism scheme mentioned above provides a reasonable assumption for the interference process. Overall, the box model predicts the interference of PAN measurement caused by NO and $NO_2$ in the ANs channel. Next, the correction factors (*C2*) of different cases under various amounts of NO, $NO_2$ and PAN added in the ANs channel are simulated to form the second look-up table.

$$[NO_2\_380] = \frac{[PNs\_real]}{C2} + \frac{[ANs\_real]}{C3} \qquad (10)$$

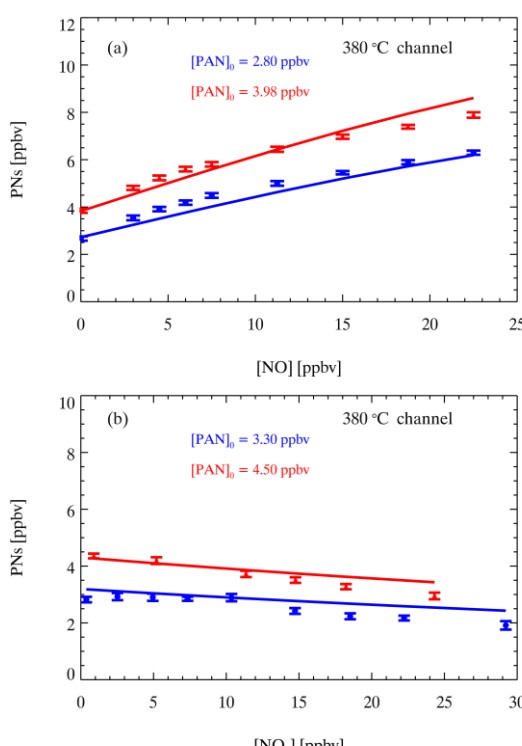

Figure 10. Simulated (lines) and measured (points) differences between the $NO_2$ signal in the ANs channel and reference channel for different PAN sources with different amounts of NO (a) and $NO_2$ (b). The error bars show one standard deviation.

The raw ONs mixing ratio ([$NO_2\_380$]) is determined by the difference between the ANs channel and reference channel based on the 'SPEC' method. According to Eq. 10, it is necessary to quantify the interference caused by ambient NO/$NO_2$ for ANs measurements. Due to the lack of ANs sources, we use box models to study the interferences by setting MeN as the representative ANs. Methyl nitrate is pyrolyzed to produce $NO_2$ and $CH_3O$ in the ANs channel (R16), and $CH_3O$ is an important intermediate product of the reactions of PAs (R10-R12). Therefore, the mechanism scheme regarding PAN applies to interference simulations of NOx for ANs measurements. Similarly, the different cases under various NO, $NO_2$ and MeN additions were simulated to form the third look-up table for the corrections of ANs measurements in the ANs channel. According to Eq. 10, the raw concentration of ANs is the

difference between the raw ONs measurements ([NO$_2$_380]) and the signal contribution of PNs ([PNs_real]/$C2$) in the ANs channel. The correction factor ($C3$) is determined by the third look-up table, and then the result is multiplied [ANs] to obtain the corrected ANs mixing ratio ([ANs_C]) by Eq. 11.

$$[ANs\_real] = \left( [NO_2\_380] - \frac{[PNs\_real]}{C2} \right) \times C3 \quad (11)$$

In addition to the interference mentioned above, other nitrogen compounds may undergo pyrolysis to generate NO$_2$ in the heated channels, such as N$_2$O$_5$ and ClNO$_2$ (Li et al., 2018; Thaler et al., 2011; Wang et al., 2017a; Womack et al., 2017), which may be a source of uncertainty for measurements of organic nitrates at night and early morning. Interferences can be extracted if the simultaneous measurements of N$_2$O$_5$ and ClNO$_2$ are available.

## 4.2 Uncertainty and detection limit

The uncertainty of the CEAS measurement of NO$_2$ is contributed by the absorption cross-section, mirror reflectivity, effective cavity length, and spectral fitting. The absorption cross-section of NO$_2$ is taken from Voigt et al. (2002), whose uncertainty is approximately 4% (Voigt et al., 2002); the uncertainty of mirror reflectivity is approximately 5%, determined by the error of the scattering cross-section of N$_2$; the uncertainty of effective cavity length is approximately 4.5%; and the uncertainty of spectral fitting when omitting the cross-section of glyoxal is 4%. According to Gaussian error propagation, the associated uncertainty of the ambient NO$_2$ measurement is ±9% based on the above parameters. The precision of the CEAS can be assessed by the Allan deviation and standard deviation (Duan et al., 2018; Langridge et al., 2008; Wang et al., 2017a). Fig. 11 shows the variance analysis of 21077 continuously measured N$_2$ spectra when the cavity was filled with N$_2$ under purge. The integration time was 3 s, and the sampling time was 6 s, as every two spectra were averaged before saving. The first 100 N$_2$ spectra collected were averaged as $I_0$, and all spectra were analyzed based on $I_0$. The data set was divided into 300 gradients for Gaussian fitting, and 1$\sigma$ was 97 pptv, as shown in Fig. 11(a). The 21077 N$_2$ spectra mentioned above were averaged at different time intervals (from 6 s to 11400 s), and then the Allan deviations at different time intervals were calculated. As shown in Fig. 11(b), the Allan deviation decreases as the sampling time increases when the sampling time is smaller than 1300 s, and the minimum is 5 pptv. When the sampling time is 6 s, the Allan variance can reach 90 pptv, close to 1$\sigma$. ANs and PNs are detected by the same CEAS system and calculated by the dynamic $I_0$ ('SPEC') method; therefore, their precision is identical to the NO$_2$ measurement. The uncertainty of [ANs] and [PNs] mainly comes from spectral fitting to derive the concentration of NO$_2$ and the interference correction in heated channels, which should be larger than 9%.

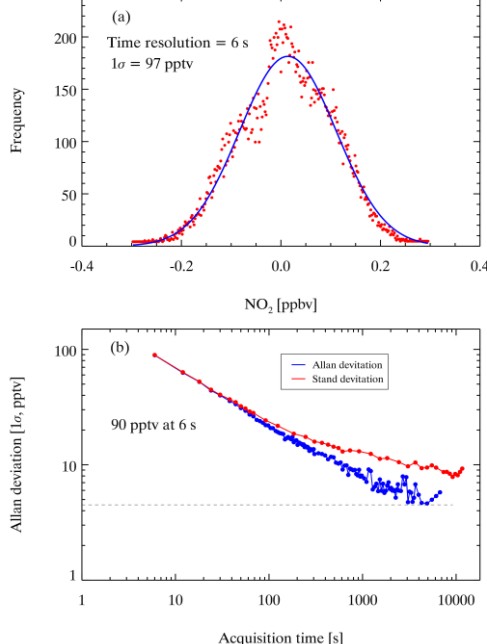


Figure 11. Instrument performance with different integration times. Panel (a) The standard deviation of the measurements of NO$_2$ with
a 6 s integration time. Panel (b) Allan deviation plots for measurements of NO$_2$ with a 6 s integration time.
As summarized in Table 2, there are several typical technologies to measure organic nitrates based on the thermal
dissociation method. TD-LIF is the pioneer to determine organic nitrates by measuring NO$_2$ produced through
pyrolysis (Day et al., 2002), and the technology has been developed well and deployed in considerable campaigns
(Di Carlo et al., 2013; Farmer et al., 2006). TD-LIF has a high time resolution and low detection limit, but the
determination of NO$_2$ has to rely on extra calibration. TD-CIMS has a similar limitation as TD-LIF, and the method
can measure some individual species of PNs, which need corresponding standards to be calibrated one by one
(Slusher et al., 2004). CRDS, CAPS and CEAS are all cavity-enhanced techniques with high sensitivity and time
resolution, of which CRDS and CAPS have been applied to detect NO$_2$ after ON pyrolysis. Specifically, in this study,
the ONs and PNs are determined directly through broadband absorption measurement by CEAS, which can avoid the
uncertainty caused by multiple spectral fitting and subsequent differential calculations. Overall, TD-CEAS has a
detection capacity similar to that of TD-LIF and others. Recently, PERCA-CRDS was developed to indirectly
determine PNs by measuring NO$_2$ through chemical amplification, which also showed high sensitivity, but the
technology for atmospheric measurements needs to be studied further.
Table 2. Typical thermal dissociation methods to measure organic nitrates.

| Method | Targets | Time resolution | Detection limit | Accuracy | Reference |
|---|---|---|---|---|---|
| TD-LIF | ANs, PNs | 10 s | 90 pptv | 10-15% | Day et al., 2002 |
| TD-LIF | ANs, PNs | 1 s | 18.4, 28.1 pptv | 22%, 34% | Di Carlo et al., 2013 |
| TD-CIMS | PAN, PPN | 1 s | 7, 4 pptv | 20% | Slusher et al., 2004 |
| TD-CRDS | ANs, PNs | 1 s | 100 pptv | 6% | Paul et al., 2009 |
| TD-CAPS | PNs, ONs | 2 min | 7 pptv | N.A. | Sadanaga et al., 2016 |
| TD-CRDS | ANs, PNs | 1 s | 28 pptv | 6%+20 pptv | Thieser et al., 2016 |
| TD-CRDS | ANs, PNs | 1 s | 59, 94 pptv | 8%+10 pptv | Sobanski et al., 2016 |
| PERCA-CRDS | PNs, PAN | 1 s | 6.8, 2.6 pptv | 13% | Taha et al., 2018 |
| TD-CEAS | ANs, PNs | 6 s | 90 pptv | 9% | This work |


## 4.3 Performance in field observations

TD-CEAS deployed the first field observations in Xinjin County, Chengdu, China, in 2019, referred to as the CHOOSE campaign (Yang et al., 2020). As shown in Fig. S8, there is a residential area 5 km northwest of the site; the surrounding area is lush with trees and is close to a forest park and a national wetland park; and there is an industrial park approximately 12 km to the west and 8 km to the south. During the CHOOSE campaign, TD-CEAS was deployed in a container. The sampling inlet protruded from the container top and was supported by a bracket with a height of 4 m above the ground. We determined the raw data of PNs and ONs during the observation period, and then the raw data were corrected as mentioned above. Fig. 12(a) shows the time series of raw data and the corrected data of PNs from August $9^{th}$ to August $18^{th}$ during the CHOOSE campaign, and Fig. 12(b) shows the time series of ANs measurements before and after correction. The correction factors are shown in Fig. S9. The value of $C1$ was generally greater than 1.0 (except during the morning), suggesting that the role of $NO_2$ was more significant than that of NO at this site. The tendency of $C2$ and $C3$ was consistent with $C1$ during measurements, but the daily changes of $C2$ and $C3$ were relatively smaller as the sensitivity of interferences in the ANs channel decreased, as mentioned above. Fig. 12(c) shows that $NO_2$ constantly increased at night, reaching a peak near the early morning and maintaining a high value to approximately 11 a.m. High mixing ratios of PNs were observed during the measurement, and the diurnal variation of PNs was clear. The peak of ANs appeared in the noontime and several hours before that of PNs. However, when the ambient $NO_2$ changed drastically at night during the campaign, the difference in $NO_2$ between adjacent measurement phases in a cycle was great, resulting in unfeasible measurements (Fig. S10). Simultaneous measurements showed that the $N_2O_5$ mixing ratio during nighttime was low and zero during the daytime. Therefore, the interferences of $N_2O_5$ were negligible for the ONs measurements during the daytime during the CHOOSE campaign. Nevertheless, the observed ANs may have been subject to the interference from $ClNO_2$.

A photolytic conversion chemiluminescence detector (PC-CLD) was used to measure NO and $NO_2$ during the campaign. The time series of $NO_2$ measured by the TD-CEAS and PL-CLD with a 5 min average is shown in Fig. 13(a), which were from August $9^{th}$ to August $16^{th}$. The trend of $NO_2$ measured by the two instruments agrees well, but the results of the PL-CLD are higher when the mixing ratio of $NO_2$ was low at noon. Fig. 13(b) shows that the correlation coefficient of the $NO_2$ concentration measured by the two instruments is 0.99. The results of the TD-CEAS are slightly higher than the results of the PL-CLD, as the slope is 0.95, which is reasonable when considering instrument uncertainties. The time series of PNs (TD-CEAS) and PAN (GC-ECD) is shown in Fig. 13(c), and the trends are relatively consistent, but the results of PNs are higher than the results of PANs measured by the GC-ECD, especially at noon. This result is reasonable since PAN concentration is the highest but is not equal to the total concentration of PNs. The correlation between the two instruments is good, as the correction coefficient is up to 0.85 (Fig. 13(d)), suggesting our instrument's feasibility in PNs measurement.

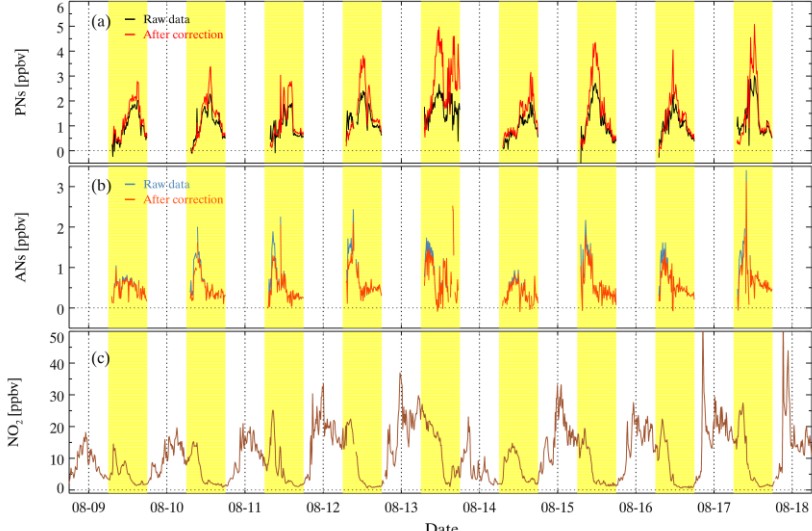


Figure 12. Time series of the observed mixing ratios of PNs, ANs and NO$_2$ during ozone pollution from the CHOOSE campaign in 2019.
The yellow regions indicate the daytime period. (a) The black lines represent the raw measurements of PNs, and the red lines are the
corrected measurements of PNs according to the look-up table. (b) The blue lines represent the raw measurements of ANs, and the orange
lines are the corrected measurements of ANs according to the look-up tables. (c) Measurements of NO$_2$ in the reference channel.

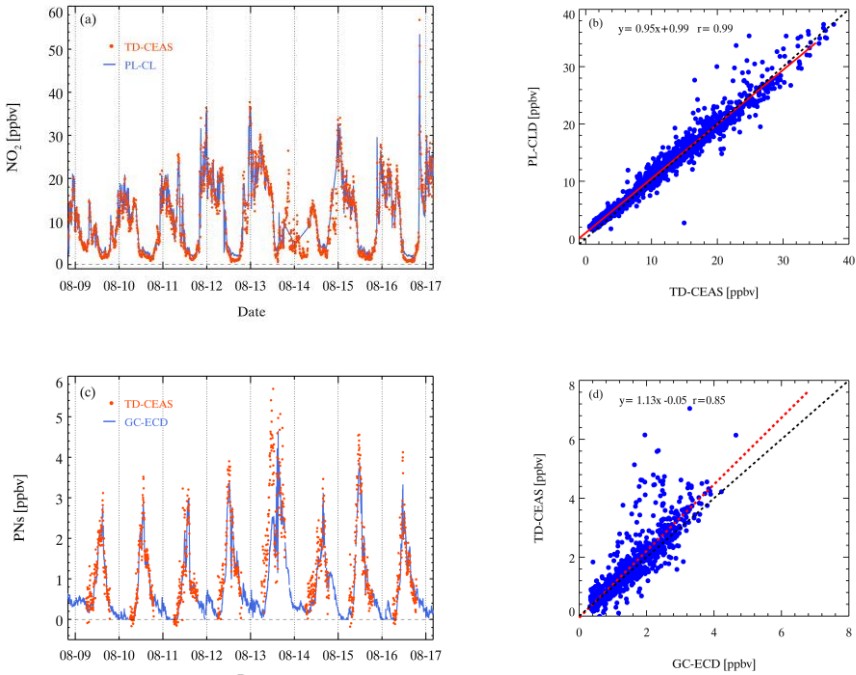


Figure 13. Comparison of the TD-CEAS and PL-CLD in the CHOOSE campaign. Panel (a) shows the time series of NO$_2$ measurements.
Orange points represent the results from the TD-CEAS, and the blue line represents the results from the PL-CLD. Panel (b) shows the
NO$_2$ correlation between the two instruments. Comparison of the PNs measured by the TD-CEAS and the PAN measured by the GC-
ECD. Panel (c) shows the time series of PNs and PAN, orange points represent the results from the TD-CEAS, and blue line represents
the results from the GC-ECD. Panel (d) shows the PNs correlation of the two instruments.

# 5. Conclusions and outlook

We developed a new and robust TD-CEAS instrument to measure PNs, ANs, and $NO_2$ in the atmosphere with high accuracy and sensitivity. The advantage of this equipment is that only one detector is used for measuring $NO_2$ at 435 - 455 nm, which reduces the potential for cross-interference caused by multiple detectors and has a lower cost and simpler operation. One measurement cycle of the instrument is 3 min, with an LOD of 97 pptv ($1\sigma$) at 6 s. The short cycle period meets the atmospheric lifetime requirements of organic nitrates and $NO_2$ in general. The measurement interferences are characterized under different NO, $NO_2$, and organic nitrates (PAN or MeN) by laboratory experiments and model simulations. A look-up table method was established to correct the PNs and ANs concentrations.

The instrument was first deployed for field measurements in Chengdu, China, and the PNs measured by the TD-CEAS showed good consistency with PAN measured by a GC-ECD during the daytime. However, when the ambient $NO_2$ in the sampled air masses changes drastically, there will be great errors for the measurement of ANs and PNs, as the $NO_2$ mixing ratio between adjacent measurement phases in a cycle will be definitely different. Adding another $NO_2$-CEAS in parallel in the instrument for continuous $NO_2$ measurement will avoid this limitation. In addition, the observed PNs and ANs may be subject to interference from other reactive nitrogen species, such as $N_2O_5$ and $ClNO_2$, which can be corrected with simultaneous measurements in the future. Overall, this instrument is suitable for measuring $NO_2$, PNs, and ANs in chamber studies or ambient measurements with relatively stable air masses free of intensive NOx emissions. We highlight the impact of interference reactions in heated channels for accurately measuring PNs and ANs. Although the look-up table can correct the interferences, the best way to reduce them is to quench $RO_2$ during the sampling process by improving the instrument design, such as by increasing the wall loss of $RO_2$ in the heated channel.

**Data availability.** The datasets used in this study are available from the corresponding author upon request (k.lu@pku.edu.cn).

**Author contributions.** K.D.L. and H.C.W. designed the study. C.M. L and H.C.W. set up and characterized the instrument, analyzed the data and wrote the paper with input from K.D.L. All authors contributed to the field measurements and discussed and improved the paper.

**Competing interests**. The authors declare that they have no conflicts of interest.

**Acknowledgments**. This project is supported by the Beijing Municipal Natural Science Foundation for Distinguished Young Scholars (JQ19031); the special fund of the State Key Joint Laboratory of Environment Simulation and Pollution Control (21K02ESPCP); the National Natural Science Foundation of China (21976006) and the National Research Program for Key Issue in Air Pollution Control (DQGG0103-01).

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
