# Peer review of "Thermal dissociation cavity-enhanced absorption spectrometer for measuring NO2,"

_Atmospheric Measurement Techniques, 2020_

## Author Comment (AC1)

**Response to Referee #1**

All of the line numbers refer to Manuscript ID: amt-2020-520.

We thank the referee's valuable comments and sugestions, we response the comments points to points, revised the manuscript carefully, and polished the language throughout the manuscript. As detailed below, the referee's comments are shown as italicized font, our response is in orange, new or modified text is in blue.

The authors report on a new dissociation cavity enhanced absorption spectrometer for quantification of NO2, RO2NO2 and RONO2 in the atmosphere. The instrument relies on cavity-enhanced absorption spectroscopy to quantify NO2 and NO2 generated from organic nitrates by sampling through a heated inlet similar to what has been described by others (Thieser et al., 2016; Paul et al., 2009; Keehan et al., 2020; Chen et al., 2017; Wooldridge et al., 2010; Sadanaga et al., 2016; Di Carlo et al., 2013). Inlet characterization and sample field data are presented.

Overall, this is a well written manuscript suitable for publication for AMT after my comments below have been addressed by the authors. Thanks for the referee's positive comments.

Major comments:

1. Considering the large body of existing TD literature, a table comparing this new instrument to existing methods and a discussion of the differences, advantages and disadvantages should be added to the paper.

Thanks for the suggestion. The table comparing our instrument to existing methods and discussions were added in the revised manuscript in Section 4.2.

As summarized in Table 2, there are several typical technologies to measure organic nitrates based on the thermal dissociation method. TD-LIF is the pioneer to determine organic nitrates by measuring NO2 produced through pyrolysis (Day et al., 2002), and the technology has been developed well and deployed in considerable campaigns (Di Carlo et al., 2013; Farmer et al., 2006). TD-LIF has a high time resolution and low detection limit, but the determination of NO2 has to rely on extra calibration. TD-CIMS has a similar limitation as TD-LIF, and the method can measure some individual species of PNs, which need corresponding standards to be calibrated one by one (Slusher et al., 2004). CRDS, CAPS and CEAS are all cavity-enhanced techniques with high sensitivity and time resolution, of which CRDS and CAPS have been applied to detect NO2 after ON pyrolysis. Specifically, in this study, the ONs and PNs are determined directly through broadband absorption measurement by CEAS, which can avoid the uncertainty caused by multiple spectral fitting and subsequent differential calculations. Overall,

TD-CEAS has a detection capacity similar to that of TD-LIF and others. Recently, PERCA-CRDS was developed to indirectly determine PNs by measuring  $NO_2$  through chemical amplification, which also showed high sensitivity, but the technology for atmospheric measurements needs to be studied further.

| Method     | Targets  | Time resolution | Detection limit | Accuracy   | Reference               |
|------------|----------|-----------------|-----------------|------------|-------------------------|
| TD-LIF     | ANs, PNs | 10 s            | 90 pptv         | 10-15%     | (Day et al., 2002)      |
| TD-LIF     | ANs, PNs | 1 s             | 18.4, 28.1 pptv | 22%, 34%   | (Di Carlo et al., 2013) |
| TD-CIMS    | PAN, PPN | 1 s             | 7, 4 pptv       | 20%        | (Slusher et al., 2004)  |
| TD-CRDS    | ANs, PNs | 1 s             | 100 pptv        | 6%         | (Paul et al., 2009)     |
| TD-CAPS    | PNs, ONs | 2 min           | 7 pptv          | N.A.       | (Sadanaga et al.,       |
|            |          |                 |                 |            | 2016)                   |
| TD-CRDS    | ANs, PNs | 1 s             | 28 pptv         | 6%+20 pptv | (Thieser et al., 2016)  |
| TD-CRDS    | ANs, PNs | 1 s             | 59, 94 pptv     | 8%+10 pptv | (Sobanski et al., 2016) |
| PERCA-CRDS | PNs, PAN | 1 s             | 6.8, 2.6 pptv   | 13%        | (Taha et al., 2018)     |
| TD-CEAS    | ANs, PNs | 6 s             | 90 pptv         | 9%         | This work               |

Table 2. Typical thermal dissociation methods to measure organic nitrates.

Specific comments:

- Title replace 'detecting' with 'measuring' or 'quantification of' (the instrument does not merely detect the presence of PN and AN after all).
   The title is modified as: "Thermal dissociation cavity enhanced absorption spectrometer for measuring NO2, RO2NO2 and RONO2 in the atmosphere."
- *3. line 102. Since the instrument samples through a PTFE filter, the Mie scattering component should be zero.*

The Eq.1 is a standard euquiton of CEAS to show the different components of the extinction coefficient. We believe there are still some particulates that can enter the cavity despite of the usage of PTFE filter, since the filter efficiency can not reach to 100%. However, the contribution of Mie scattering and Rayleigh scattering is eliminated by polynomial fitting through DOASIS.

4. line 110. Please comment on the precision of the output power of the stabilized source.

The LED optical output power is about 1850 mW. We continuously measured N2 spectral to dectect the stability of the light source. The figure shows the signal distribution of 1000 N2 spectral, and the uncertainty of the signals is

Line 113-115, revised as: "The core of the light source module is a single-color LED (M450D3, Thorlabs, Newton, NJ, USA), which emits more than 1850 mW optical power at approximately 450 nm with a full width at half maximum (FWHM) of 18 nm."

5. line 119. reported by whom? ATF?

Yes, it is. The values of reflectivity of the high-reflectivity mirror are cited from the report of ATF.

Here we revised this sentence as follow: "The reflectivity of HR mimmors (CRD450-1025-100, Advanced thin films, CO, USA) is reported by the manufacturer to be greater than 0.9999 (440 - 460 nm) with a radius curvature of 1.0 m and a diameter of 25.4 mm."

6. line 155 please state the manufacturer and internal surface of the T-shaped solenoid valves.

The text is revised as follows: "..., a time relay is used to periodically control the three T-shaped solenoid valves (71335SN2KVJ1, Parker Hannfin, USA), and the internal surface of the T-shaped solenoid valves is stainless steel."

7. line 160. Hg lamps tend to "run hot" which can affect the output of PAN, which is prone to thermal dissociation. Has the composition and purity of the PAN source been evaluated?

Furthermore, what was the concentration or mixing ratio of acetone used?

We used a PAN source to do the experiments after the temperature of the Hg lamp in the PAN source stays stable at 39 °C, so the output of PAN source is stable. The PAN source used here is a calibration part of GC-ECD instrument, and the source level has been checked by GC-ECD. As the following figure shows, the PAN source work stably, checked by GC-ECD, with an average of  $5.18 \pm 0.03$  ppbv. The mixing ratio of acetone standard used here is 500 ppmv.

"The source was used for the laboratory experiments after the temperature of the

Hg lamp stablised at 39.0 °C, and the source level and stability were double-checked by a GC-ECD instrument."

8. line 166 which requirements?

The requirements refer to the need to use some typical mixing ratio of  $NO_2/NO$  during instrument characterization and interference correction, so the text is revised as follows: "..., and outputted well-mixed gases by diluting NO or  $NO_2$  with zero air according to the requirement of studying the potential interference caused by ambient NO and  $NO_2$ ."

9. line 182. Please describe how  $N_2$  and He were delivered (sampled from the tip of the inlet, or statically).

 $N_2$  and He were delivered through the purge flow injection lines, so the text is revised as follows: "The spectra of pure  $N_2$  (>0.99999) or He (>0.99999) filling the cavity through the purge lines, are collected to calibrate the mirror reflectivity, ..."

10. line 226. Please justify omitting glyoxal from the fit and estimate the uncertainty introduced. Figure 4 suggests that the contribution of glyoxal was small but that may not always be the case. What (if anything) happens to the NO2 retrieval when glyoxal is included in the fit?

We agree with that the including of glyoxal retrieval would affect the determination of NO2. We compare the two cases with and without the glyoxal retrieval during the spectral fitting. The comparison shows the inclusion of glyoxal bring bigger fitting residual, which is consistent with the result of recent study (Liu et al., 2019). While the difference of retrived NO2 concentration between the two fitting method is small. As shown in the following figure, the relative difference of derived NO2 concentration suggests the uncertainty by omitting  $\sigma$ (glyoxal) from spectral fitting is about 4%.

Line 240 revised as: "...but here, we do not take glyoxal absorption into

consideration in spectal fitting. The inclusion of glyoxal in the spectral fitting would enlarge the fitting residual. Our field measurement showed that the uncertainty caused by excluding glyoxal fitting was approximately 4% (Fig. S2)."

Figure S2. An example of the comparison about the fitting results with considering the cross section of glyoxal or without. Panels show the case on August 17, 2019 and August 18, 2019 during CHOOSE campaign. Panel (a) shows the time series of NO2 under two different conditions. Panel (b) shows the time serie of glyoxal under Case 2. Panel (c) shows the concentration ratio of NO2 under two different cases.

**11. line 236 - Figure 4. Please state the uncertainty of the NO2 mixing ratios (16.2±? ppbv).**

The uncertainty caused by the residual of spectral fitting are added in Figure 4 (16.2  $\pm$  0.1 ppbv, 1.8  $\pm$  0.1 ppbv), and revised accordingly in the manuscript.

---

## Author Response (AR1)

**Response to Referees**

All of the line numbers refer to Manuscript ID: amt-2020-520.

We thank the referees' valuable comments and sugestions, we response the comments points to points, revised the manuscript carefully, and polished the language throughout the manuscript. As detailed below, the referees' comments are shown as italicized font, our response is in orange, new or modified text is in blue.

*Referee # 1*

*The authors report on a new dissociation cavity enhanced absorption spectrometer for quantification of $NO_2$, $RO_2NO_2$ and $RONO_2$ in the atmosphere. The instrument relies on cavity-enhanced absorption spectroscopy to quantify $NO_2$ and $NO_2$ generated from organic nitrates by sampling through a heated inlet similar to what has been described by others (Thieser et al., 2016; Paul et al., 2009; Keehan et al., 2020; Chen et al., 2017; Wooldridge et al., 2010; Sadanaga et al., 2016; Di Carlo et al., 2013). Inlet characterization and sample field data are presented.*

*Overall, this is a well written manuscript suitable for publication for AMT after my comments below have been addressed by the authors.*

Thanks for the referee's positive comments.

 *Major comments:*

1. *Considering the large body of existing TD literature, a table comparing this new instrument to existing methods and a discussion of the differences, advantages and disadvantages should be added to the paper.*

    Thanks for the suggestion. The table comparing our instrument to existing methods and discussions were added in the revised manuscript in Section 4.2.

    As summarized in Table 2, there are several typical technologies to measure organic nitrates based on the thermal dissociation method. TD-LIF is the pioneer to determine organic nitrates by measuring $NO_2$ produced through pyrolysis (Day et al., 2002), and the technology has been developed well and deployed in considerable campaigns (Di Carlo et al., 2013; Farmer et al., 2006). TD-LIF has a high time resolution and low detection limit, but the determination of $NO_2$ has to rely on extra calibration. TD-CIMS has a similar limitation as TD-LIF, and the method can measure some individual species of PNs, which need corresponding standards to be calibrated one by one (Slusher et al., 2004). CRDS, CAPS and CEAS are all cavity-enhanced techniques with high sensitivity and time resolution, of which CRDS and CAPS have been applied to detect $NO_2$ after ON pyrolysis. Specifically, in this study, the ONs and PNs are determined directly through broadband absorption measurement by CEAS, which can avoid the uncertainty

caused by multiple spectral fitting and subsequent differential calculations. Overall, TD-CEAS has a detection capacity similar to that of TD-LIF and others. Recently, PERCA-CRDS was developed to indirectly determine PNs by measuring $NO_2$ through chemical amplification, which also showed high sensitivity, but the technology for atmospheric measurements needs to be studied further.

Table 2. Typical thermal dissociation methods to measure organic nitrates.

| Method | Targets | Time resolution | Detection limit | Accuracy | Reference |
|---|---|---|---|---|---|
| TD-LIF | ANs, PNs | 10 s | 90 pptv | 10-15% | Day et al., 2002 |
| TD-LIF | ANs, PNs | 1 s | 18.4, 28.1 pptv | 22%, 34% | Di Carlo et al., 2013 |
| TD-CIMS | PAN, PPN | 1 s | 7, 4 pptv | 20% | Slusher et al., 2004 |
| TD-CRDS | ANs, PNs | 1 s | 100 pptv | 6% | Paul et al., 2009 |
| TD-CAPS | PNs, ONs | 2 min | 7 pptv | N.A. | Sadanaga et al., 2016 |
| TD-CRDS | ANs, PNs | 1 s | 28 pptv | 6%+20 pptv | Thieser et al., 2016 |
| TD-CRDS | ANs, PNs | 1 s | 59, 94 pptv | 8%+10 pptv | Sobanski et al., 2016 |
| PERCA-CRDS | PNs, PAN | 1 s | 6.8, 2.6 pptv | 13% | Taha et al., 2018 |
| TD-CEAS | ANs, PNs | 6 s | 90 pptv | 9% | This work |

*Specific comments:*

2. *Title - replace 'detecting' with 'measuring' or 'quantification of' (the instrument does not merely detect the presence of PN and AN after all).*

   The title is modified as: "Thermal dissociation cavity enhanced absorption spectrometer for measuring $NO_2$, $RO_2NO_2$ and $RONO_2$ in the atmosphere."

3. *line 102. Since the instrument samples through a PTFE filter, the Mie scattering component should be zero.*

   The Eq.1 is a standard euqaiton of CEAS to show the different components of the extinction coefficient. We believe there are still some particulates that can enter the cavity despite of the usage of PTFE filter, since the filter efficiency can not reach to 100%. However, the contribution of Mie scattering and Rayleigh scattering is eliminated by polynomial fitting through DOASIS.

4. *line 110. Please comment on the precision of the output power of the stabilized source.*

   The LED optical output power is about 1850 mW. We continuously measured $N_2$ spectral to dectect the stability of the light source. The figure shows the signal distribution of 1000 $N_2$ spectral, and the uncertainty of the signals is <0.5%. For example, the mean of signals at 445 nm is 30636 counts, and the $3\sigma$ is equal to 111 counts.

[Figure]

Line 113-115, revised as: "The core of the light source module is a single-color LED (M450D3, Thorlabs, Newton, NJ, USA), which emits more than 1850 mW optical power at approximately 450 nm with a full width at half maximum (FWHM) of 18 nm."

5. *line 119. reported by whom? ATF?*

Yes, it is. The values of reflectivity of the high-reflectivity mirror are cited from the report of ATF.

Here we revised this sentence as follow: "The reflectivity of HR mimmors (CRD450-1025-100, Advanced thin films, CO, USA) is reported by the manufacturer to be greater than 0.9999 (440 - 460 nm) with a radius curvature of 1.0 m and a diameter of 25.4 mm."

6. *line 155 please state the manufacturer and internal surface of the T-shaped solenoid valves.*

The text is revised as follows: "…, a time relay is used to periodically control the three T-shaped solenoid valves (71335SN2KVJ1, Parker Hannfin, USA), and the internal surface of the T-shaped solenoid valves is stainless steel."

7. *line 160. Hg lamps tend to "run hot" which can affect the output of PAN, which is prone to thermal dissociation. Has the composition and purity of the PAN source been evaluated?*
*Furthermore, what was the concentration or mixing ratio of acetone used?*

We used a PAN source to do the experiments after the temperature of the Hg lamp in the PAN source stays stable at 39 °C, so the output of PAN source is stable. The PAN source used here is a calibration part of GC-ECD instrument, and the source level has been checked by GC-ECD. As the following figure shows, the PAN source work stably, checked by GC-ECD, with an average of $5.18 \pm 0.03$ ppbv. The mixing ratio of acetone standard used here is 500 ppmv.

"The source was used for the laboratory experiments after the temperature of the

Hg lamp stablised at 39.0 °C, and the source level and stability were double-checked by a GC-ECD instrument."

[Figure]

8. *line 166 which requirements?*

   The requirements refer to the need to use some typical mixing ratio of $NO_2/NO$ during instrument characterization and interference correction, so the text is revised as follows: "…, and outputted well-mixed gases by diluting NO or $NO_2$ with zero air according to the requirement of studying the potential interference caused by ambient NO and $NO_2$."

9. *line 182. Please describe how $N_2$ and He were delivered (sampled from the tip of the inlet, or statically).*

   $N_2$ and He were delivered through the purge flow injection lines, so the text is revised as follows: "The spectra of pure $N_2$ (>0.99999) or He (>0.99999) filling the cavity through the purge lines, are collected to calibrate the mirror reflectivity, …"

10. *line 226. Please justify omitting glyoxal from the fit and estimate the uncertainty introduced. Figure 4 suggests that the contribution of glyoxal was small but that may not always be the case. What (if anything) happens to the $NO_2$ retrieval when glyoxal is included in the fit?*

    We agree with that the including of glyoxal retrieval would affect the determination of $NO_2$. We compare the two cases with and without the glyoxal retrieval during the spectral fitting. The comparison shows the inclusion of glyoxal bring bigger fitting residual, which is consistent with the result of recent study (Liu et al., 2019). While the difference of retrived $NO_2$ concentration between the two fitting method is small. As shown in the following figure, the relative difference of derived $NO_2$ concentration suggests the uncertainty by omitting σ(glyoxal) from spectral fitting is about 4%.

    Line 240 revised as: "…but here, we do not take glyoxal absorption into

consideration in spectal fiting. The inclusion of glyoxal in the spectral fitting would enlarge the fitting residual. Our field measurement showed that the uncertainty caused by excluding glyoxal fitting was approximately 4% (Fig. S2)."

[Figure]

Figure S2. An example of the comparison about the fitting results with considering the cross section of glyoxal or without. Panels show the case on August 17, 2019 and August 18, 2019 during CHOOSE campaign. Panel (a) shows the time series of $NO_2$ under two different conditions. Panel (b) shows the time serie of glyoxal under Case 2. Panel (c) shows the concentration ratio of $NO_2$ under two different cases.

11. *line 236 - Figure 4. Please state the uncertainty of the $NO_2$ mixing ratios (16.2±? ppbv).*

   The uncertainty caused by the residual of spectral fitting are added in Figure 4 (16.2 ± 0.1 ppbv, 1.8 ± 0.1 ppbv), and revised accordingly in the manuscript.

[Figure]

12. *line 240 - Figure 5. It seems that the timing of the inlet temperature switch was off*

*as there are blue data points at the same level as [NO₂]ambient and red data points at the same level as 180 °C.*

As the Figure 5 shows, a measurement cycle includes 3 phases whose furation is 60 s, but the CEAS may still detect the air flow from previous channel when the measurement phases just switch. Here we excluded the transition point of each phases and the two data points before and after the transition point to avoid the misundetstanding. The Figure 5 is revised as follow.

[Figure]

13. *lien 248 It is 1 spectrum, but the plural should be 2 spectra (not "spectrums").*
    Corrected accordingly.

14. *line 283. An alternative (and more likely) interpretation of the second plateau is the presence of alkyl nitrate impurity. What are the operating temperature and the output purity of the photochemical PAN source? Is [PAN]out = [NOx]in ?*

    Thank you for the comment. The operature temperature of photochemical PAN source stays 39 °C, and [PAN]out equals to $92 \pm 3\%$ of [NOx]in.

    The presence of alkyl nitrates in PAN source was once reported by previous reports (Paul et al., 2009). In this study, we can not ruled out the possibility of the alkyl nitrate impurity. But the fact that the source of [PAN]out is equal to $92 \pm 3\%$ of [NOx]in, suggesting that only a very small percentage ($\leqslant$ 8% on average), if any, of the ANs. Futhermore, Figure 7 demonstated that the second plateau can be well explained by recombination of $NO_2$ with $RO_2$, and Figure 8 showed the concentration of PAN source and wall loss rate of $RO_2$ affect the presence of the second plateau, so it is reasonable of our explatation that attribute the formation of the second plateau to recombination reaction of PAN.

    In the revised manuscript, we added the following sentences to discuss the possibility of ANs impurity to the dissociation profile of PNs.

    Line 294: "approximately 400 °C (Friedrich et al., 2020). The presence of alkyl nitrates in PAN source has been reported before by previous studies, and was regarded as the reason of the dual-plateau profile of PNs dissociation (Paul et al.,

2009). Here, we cannot ruled out the possibility of alkyl nitrate impurities. However, the source level of PAN is equal to 92 ± 3% of NOx imput, which suggests that only a very small percentage (⩽ 8% on average), if any, of the ANs."

15. *lines 283-284. The PAN dissociation temperature of 400 °C reported by Friedrich et al. is an outlier and inconsistent with every other paper on this subject. At a residence time of 142 ms (line 153) and using rate constant for unimolecular decomposition reported by (Kabir et al., 2014), PAN is predicted to be 99% dissociated at a temperature of "only" 127 °C.*

    We agree with the comment. As our model result shown in the following figure, PAN is well thermally dissociated to $NO_2$ at 180 °C in the heated tube, but the recombination of PAN not only occurrs at heated part but also at the subsqunt cooling part in the heated channel. The overall dissociation temperature of PAN depends not only on the concentration of PAN source and the wall loss rate of $RO_2$, but also on the settings of the instrument. The line 285-287 is revised as follows: "…If the PAN source is equal to 4 ppbv in the PNs channel at 180 °C, as Fig. S3 shows, PAN will first be dissociated completely, and then the PA will recombine with $NO_2$ to form PAN when the air flow passes the cooling lines."

[Figure]

    Figure S3. A simulated example of the PAN pyrolysis in the PNs channel at 180 °C if the PAN source is equal to 4 ppbv. The concentration of relative species changes with the residence time, the red line is concentration of PAN, the blue line is the concentration of PA radical, and the yellow line is the concentration of $NO_2$. The red part in the plot is the duration time when the air flow goes through the heating part of quartz tube, and the blue part is the duration time when the air flow goes through the cooling part.

16. *line 286 - how much time is there for recombination to occur?*

    According to the simulation results of the box model, when the setting temperature is 180 ℃,the duration time for the recombination to occur after the heated tube to the detector is 297 ms.

17. *line 290. CH₃O₂NO₂ more readily dissociates than PAN; under the conditions of the authors' inlet, it is predicted to be >99% dissociated at a temperature of <50 °C.*

Yes, $CH_3O_2NO_2$ more readily dissociates than PAN, but $CH_3O_2NO_2$ is one of the derivative products during the thermal dissociation of PAN. As Figure 7 shows, the presence of $CH_3O_2NO_2$ only occurs at pretty high temperature, and $CH_3O_2NO_2$ comes from the following two reasons. Firstly, the temperature first rises and then falls in the heated tube and the subsequent cooling lines, so PAN in the air flow tend to dissociate and then falling temperature promotes some other reactions, like the recombination of PAN. Secondly, the dissociation of PA accelerates in the heated channel with the temperature increasing. Therefore there are some $CH_3O_2$ as a product of the disscaition of PA, which is likely to react with $NO_2$ to produce a small amount of $CH_3O_2NO_2$ at the cooling part.

18. *line 291. The two plateaus can also be interpreted as a ~2:1 mixture of PAN and alkyl nitrates - can this be ruled out (see question for lines 160 and 283).*

Please attach the Comment and Response #13. In order to clarify the problem, other instruments which is able to measure the individual species of ONs, may be useful to test the purify of the PAN source in the future.

19. *line 294. Please state how much time there is for recombination to occur.*

As mentioned in comment 15, when the setting temperature is 180 °C, the duration time for the recombination to occur is 297 ms. The text is revised as follows: "The first plateau at 180 °C is caused by the recombination of PA and $NO_2$ after the pyrolysis of PAN, and the time for recombination from the end of the tube to the detector is 297 ms. "

20. *line 318 Caption to Figure 7 - are gray/green and PAN/NO₂ backwards?*

NO. As Figure 7 shows, the gray columns represent undissociated or recombinated PAN. Similarly, the green columns represent $NO_2$ thermal dissociated from PAN source.

21. *line 326 Filter and wall losses are small only if they the filter and wall material are made from Teflon. Please rephrase.*

Yes, the text is revised as follows: "Previous studies have shown that the filter losses and wall losses of $NO_2$, PNs and ANs are small when using Teflon tubes and Teflon filters (Paul et al., 2009; Thieser et al., 2016)."

22. *The statement is true for ANs such as methyl or ethyl nitrate; not sure the statement is true for isoprene nitrates that are prone to hydrolysis (Vasquez et al., 2020).*

This is a good question. Once formed, isoprene nitrates are prone to incorporate into aerosol where the hydrolysis of isoprene nitrates will happen. This is a vital problem for the sampling of isoprene nitrates. We do not evaluate the sampling loss of the isoprene nitrates. The wall loss is likely to be reduced if the frequency of filter changing is improved.

23. *line 426 Replace MeN with Methyl nitrate (one is not supposed to start a sentence with an abbreviation or acronym).*
Corrected accordingly.

24. *line 448. It is worthwhile noting that these molecules are important only at night and early morning hours - for this reason, the Cohen group has generally not reported AN data at those times of day.*
*Please cite (Thaler et al., 2011) for ClNO₂ and (Womack et al., 2017) for N₂O₅.*
The text has changed as : "In addition to the interference mentioned above, other nitrogen compounds may undergo pyrolysis to generate $NO_2$ in the heated channels, such as $N_2O_5$ and $ClNO_2$ (Li et al., 2018; Thaler et al., 2011; Wang et al., 2017a; Womack et al., 2017), which may be a source of uncertainty for measurements of organic nitrates at night and early morning. Interferences can be extracted if the simultaneous measurement of $N_2O_5$ and $ClNO_2$ are available."

25. *line 462/465. It is 1 spectrum, and 2 spectra (not "spectrums").*
Corrected accordingly.

26. *line 465 "as shown in Fig. 9a. The 21077 spectrums" Figure 9a does not show this information (11a perhaps?). What is meant by 21077?*
Thank you for the correction. Line 465 "as shown in Fig. 9a. The 20177 spectrums" should be "as shown in Fig. 11(a). The 20177 $N_2$ spectra". The 21077 is the number of spectra applied to calculate Allan variance and LOD of the instrument.

27. *line 467 Fig. 9b should be 11b.*
Corrected accordingly.

28. *line 500 "up to 0.99" - please state the actual value of r*
The text is revised as follows: "Fig. 13(b) shows that the correlation coefficient of the $NO_2$ concentration measured by the two instruments is 0.99."

29. *line 514 Figure 13. I am not sure what is plotted here. The GC-ECD data are labeled "PAN" on the left-hand side, but PNs on the right-hand side. Tthe TD-CEAS*

*data are labeled PNs on the left but PAN on the right.*

*In principle, the GC-ECD can observe PAN, PPN etc. and those can be summed to ΣPN. Was this done?*

Thank you for the correction, there was a label mistake. We corrected and replot it as follows.

[Figure]

**Referee # 2**

*Summary: The authors present a new thermal dissociation cavity enhanced absorption spectrometer (TD-CEAS) for measurement of $NO_2$, peroxy nitrates (PNs), and alkyl nitrates (ANs). They demonstrate, through lab tests and box model simulations, that interferences can be corrected for, and that the instrument outputs accurate measurements when compared to a chemiluminescence detector. Finally, results from a measurement campaign in Chengdu, China are presented, demonstrating its effectiveness in ambient conditions.*

*In general, this manuscript successfully demonstrates the performance of this new instrument, carefully considering the difficulties in converting PNs and ANs to $NO_2$ in a thermal dissociation oven. There are some significant grammatical/English errors throughout the manuscript, which in some cases make the details difficult to understand but these can be fixed. I would recommend publication, after the authors address some comments below, as well as editing the English.*

Thanks for the referee's positive and constructive comments. In addition to the detailed response to the referee' s comments, we have polished the language of the full text.

*General comments:*

1. *The authors sometimes refer to the ANs channel, and sometimes to ONs channel, which is confusing. This should be made clear throughout the manuscript that these are different, but related to each other.*

   The name of the 380 °C channel is uniformly named as ANs channel.

2. *Section 3.4 demonstrates that the 180 degree oven is not sufficiently hot enough to prevent recombination of PA and $NO_2$, with efficiencies ranging from 0.5 to 0.9 at 180 degrees. Are the authors correcting for this incomplete thermal dissociation in the rest of the paper? If so, this should be stated clearly. If not, this seems like a major inaccuracy of the measurement and should be addressed. Figure 9 makes it look like it isn't being made, since the intercept at x = 0 doesn't match what the legend says the input PAN concentration is.*

   In this secton, we showed that the thermal dissociation efficiency can be affected by the wall loss of PA and the level of PAN source in the 180 degree channel. This effect of incomplete pyrolysis of PAN has been well considered in the look-up table, and applied in the field campaign. The results showed in Figure 9 and 10 were not corrected, since the box simulation results shown in Fig.9 and Fig.10 were to reproduce the difference of measured $NO_2$ between two channels.

   The text in Sect. 3.4 is revised as follows: "…In addition, the interference of incomplete dissociation for PAN in the PNs channel at 180 °C is considered in the look-up table for correction, which is detailed in Sect. 4.1. "

3. *The authors should more clearly demonstrate how they convert the measured α(λ) to [$NO_2$]. Perhaps another equation would be helpful here in section 3.3, demonstrating that it is a linear fit of all the possible gas-phase absorbers in that wavelength region.*

   There is a equation in the manuscript shown as Eq.1, so the text is revised as follow: "The measured absorption coefficient (α) is processed by the DOASIS (Differential Optical Absorption Spectroscopy Intelligent System) according to Eq.1. "

4. *As the authors state on line 533, sudden changes in the ambient $NO_2$ while this instrument is measuring from the PNs and ANs would pose a significant problem. Probably this instrument is only useful when a simultaneous measurement of $NO_2$ is available. Most field campaigns do have $NO_2$ measurements, so this likely isn't a major issue, but the authors should address it anyway.*

   We agree with the comment. In generally, adding another $NO_2$-CEAS in parallel in our system for $NO_2$ mesurement would greatly helpful for the detection of ANs and

PNs.

The Sect. 5 is revised as follow: "However, when the ambient $NO_2$ in the sampled air masses changes drastically, there will be great errors for the measurement of ANs and PNs, as the $NO_2$ mixing ratio between adjacent measurement phases in a cycle will be definitely different. Adding another $NO_2$-CEAS in parallel in the instrument for continuous $NO_2$ measurement will avoid this limitation. "

5. *Many of the references have titles listed in all capital letters, which should be changed.*
   Corrected accordingly.

*Specific comments:*

6. *Line 32: "One is peroxy acyl nitrates (PANs)…". The other one is never defined. Is it the peroxy nitrates without an acyl group, as mentioned in line 34?*
   Yes, the text has been modified as follows: "The other is some peroxy nitrates without acyl groups, which are only abundant in cold regions (Roberts, 1990; Roberts et al., 1998b; Thieser et al., 2016; Wooldridge et al., 2010)."

7. *Line 33: Define PAN here, to differentiate from the more general PANs.*
   The definition of PAN has added in the text as fellows: "…, among which PPN (peroxypropionyl nitrate) and PAN (peroxyacetyl nitrate) dominate PNs with percentages of 75-90% due to their relatively high thermal stability."

8. *Line 46: "with a small branch ratio (1% - 30%)": Which reaction (R3a or R3b) is defined as the one with the 1 – 30% branching ratio, and which is the 70 – 99% reaction?*
   The reaction (R3a) is the one with the 1-30% branching ratio to form ANs. The text has been revised as fellows: "During the daytime, there is a branching reaction between $RO_2$ and NO to form ANs (R3a) with a small branch ratio (1-30%)…"

9. *Line 75: "the importance of PNs and ANs in regulating ozone formation has not been well studied [in China]": The absence of citations here implies it has not been studied at all, which is not true. Examples include: Liu 2010, Zhang 2014, and Liu 2018. Some citations should be included here.*
   Thanks for the suggestion, the citations have been included in the text as fellows: "Although many studies have examined the effect of PNs and ANs on regulating ozone formation (Chen et al., 2018; Ling et al., 2016; Liu et al., 2018; Liu et al., 2012; Liu et al., 2010; Zeng et al., 2019; Zhang et al., 2014), but the issue has not been well studied."

10. *Line 92: Fig S1 only shows the wavelength range 430 – 460 nm, so this line should be changed to match.*

Revised accordingly.

11. *Line 151: "... for the ANs and PNs channels are controlled at 180 degrees and 380 degrees, respectively". These numbers appear to be backwards, as the ANs channel was at 380, not 180 degrees.*

Yes, corrected as accordingly: "…, the heated temperatures for the ANs and PNs channels are controlled at 380 °C and 180 °C, respectively."

12. *Line 154: Presumably the solenoid valves are made of stainless steel? Do the authors expect any $NO_2$ losses on this steel?*

Yes, the solenoid values are made of stainless steel. However, there are cooling lines after the quarz tube in heated channels, the temperature of the air flow has returned to ambient temperature, so the wall loss of $NO_2$ is expected to be negligible as previous tests have demonstrated (Fuchs et al., 2009; Osthoff et al., 2006).

13. *Line 175: Define the MCM, and include a citation.*

The definition of the MCM and the citation have been added in the text: "…, and the reaction rate of these reactions is mainly taken from the Master Chemical Mechanism, MCM v3.3 (website: http://mcm.leeds.ac.uk/MCM) (Jenkin et al., 1997; Saunders et al., 2003). "

14. *Line 229: "The corresponding fitting residual is in the range of 10 x 10^-9, suggesting the system can guarantee the accuracy…". What is the meaning of this number and why does it imply the system's accuracy? Wouldn't it be better to compare the residuals between the two different fits to demonstrate they are similar in their magnitude?*

This statement was confusing. An accuracy fitting means that the residual is much smaller than the absorption of absorbers and without special structures of the absorbers. Here we revised the statement as follows: " The corresponding fitting residual, which is the difference between the measured and fitting results, is in the range of $10 \times 10^{-9}$ at 435-455 nm."

15. *Line 247: Move the "CONC" label to after "One is the differential concentration method" on line 245.*

Corrected accordingly.

16. *Line 245 – 255: In general, this paragraph is more confusing than it needs to be. You can simply state that there are two methods, one which calculates [NO₂] in each channel from equation (1) using N₂ as I₀, then subtracts [NO₂]_ambient to yield [ANs] and [PNs]. The other method uses I_ambient as I₀ to first derive a corrected α(λ), and then uses this to calculate [PNs] and [ANs]. I do not think that equations 3 – 8 are necessary.*

Thank you for the suggestion, we think the description of this paragraph is detailed, which maybe more friendly for readers who are not in this field to understand it. In order to state more clearly, the statement is revised as follows: " There are two methods to determine the mixing ratio of ONs and PNs. One is the differential concentration method ('CONC'), as shown in Eqs. 3-6, the $I_0$ is fixed during data analysis by using the $N_2$ spectrum: $I_{TD380}$ and $I_{TD180}$ are the spectra obtained when the CEAS detects ANs channel and PNs channel, respectively; $I_{N_2}$ is the $N_2$ spectrum obtained when the cavity is filled with $N_2$ (>0.99999); $\alpha_{TD380}$ and $\alpha_{TD180}$ are absorption coefficients when setting $I_{N_2}$ as I₀, and setting $I_{TD380}$ or $I_{TD180}$ as I, respectively; and after deleting the abnormal points caused by measurement phase switching, [ONs] is obtained by subtracting [NO₂]$_{TD380}$ from the average of [NO₂]$_{REF}$, and [PNs] is obtained by subtracting [NO₂]$_{TD180}$ from the average of [NO₂]$_{REF}$ . The other method is the differential absorption method ('SPEC'), by using the dynamic background spectrum method for spectral fitting (Eqs.7-8): $I_{REF}$ is the spectrum obtained at the reference channel; ONs can be retrieved based on $I_{TD380}$ and $I_{REF}$; PNs can be retrieved by $I_{TD180}$ and $I_{REF}$."

17. *Line 254 and elsewhere: The "SPEC" method is often misspelled as "SEPC".*
Corrected accordingly.

18. *Line 271 – 274: This is helpful information about why two different oven setpoints will yield PNs and ANs separately. It should be moved to earlier in the manuscript, perhaps in the introduction.*
The line 271-274 are moved to the introduction section.

19. *Lines 282: "platform" should be replaced with "plateau".*
Corrected accordingly.

20. *Line 345: These interferences of a few percent, while not large, are still non-negligible. Are the measurements being corrected for these interferences? If so, that should be stated clearly.*
As shown in the manuscript, the interferences for ANs channel and PNs channel are within 3% in a typical case on an ozone pollution day, which is smaller than the

uncertainty of $NO_2$ measuremnts. Therefore, the interference is ignored in the measurement correction.

The text is revised as follow: " The interferences are within 3% in the typical case, which is smaller than the uncertainty of the $NO_2$ measuremnts. Therefore, the interference is ignored in the measurement correction."

21. *Line 366: "as described above" should be "as described below"*
Corrected accordingly.

22. *Line 367: To stay consistent with previous sentence, replace "RO$_2$" with "PA"*
Corrected accordingly.

23. *Lines 426 – 441 and equations 9 – 12: This is another example of a paragraph that is much more confusing than it needs to be. It seems that you could just say that to accurately measure ANs, you must first measure PNs in the 180 degree channel, apply a corrective factor based on the first look-up table, then subtract this from the raw ANs channel, then apply a second corrective factor based on the second look-up table. The way the authors have written it, with many new parameters such as [PNs_C] is just more confusing.*

Thank you for the suggestion. In order to explain the correction more clearly, the text are revised totally as shown in Sect. 4.1 according to the following equations.

"…We derived the PNs corrected by C1 and [NO$_2$_180] as mentioned above. To determine the corrected concentrations of ANs, as Eqs. 10-11 shows, we need to apply another correction factor (C2) to determine the contribution of PNs to ONs, in which this is subtracted from the raw ONs measurement ([NO$_2$_380]), and finally the third correction factor (C3) is applied to obtain corrected ANs measurements. …"

$$[\text{PNs\_real}] = [\text{NO}_2\_180] \times C1 \tag{9}$$

$$[\text{NO}_2\_380] = \frac{[\text{PNs\_real}]}{C2} + \frac{[\text{ANs\_real}]}{C3} \tag{10}$$

$$[\text{ANs\_real}] = \left([\text{NO}_2\_380] - \frac{[\text{PNs\_real}]}{C2}\right) \times C3 \tag{11}$$

24. *Line 472: "... the interference in the heated channels, which should be larger than 8%". Where does this number come from?*

The number comes from the uncertainty of $NO_2$ measurement by CEAS as mentioned above.

The text is revised as follows: "The uncertainty of [ANs] and [PNs] mainly comes from spectral fitting to derive the concentration of $NO_2$ and the interference

correction in heated channels, which should be larger than 9%."

25. *Line 484 – 487: This is repeating how the corrections are made, and was already stated in the previous section, so it doesn't need to be repeated here. Doing so implies that the technique is different here.*
The text is revised accordingly as fellows: "We determined the raw data of PNs and ONs during the observation period, and then the raw data were corrected as mentioned above. "

26. *Line 535 – 538: These lines are introducing new information to the analysis, and should be included in the results and discussion section instead of the conclusions section.*
Yes, those new information are included in Sect. 4.3: "However, when the ambient $NO_2$ changed drastically at night during the campaign, the background $NO_2$ level (REF channel) between adjacent measurement phases in a cycle was great, resulting in the unfeasible measurements (Fig. S10). The simultaneous measurements show that the $N_2O_5$ mixing ratio during nighttime is low and zero during the daytime. Therefore, the interferences of the $N_2O_5$ were negligible for the ONs measurements during the daytime during the CHOOSE campaign. Nevertheless, the observed ANs may be subject to the inferference from $ClNO_2$. "

27. *Figure 2: Zoom in on the left-hand axis which shows reflectivity. It is difficult to see the full range of R.*
Thank you for pointing out the issue. The Figure 2 is revised as fellows.

[Figure]

28. *Figure 3: Why do the authors expect the d_eff / L vs flow rate plot to be linear? A linear fit implies that at the intercept, where flow rate = 0, then d_eff / L will be 0.79, when in fact, d_eff / L should approach 0 as the flow rate decreases to 0. On the other end, as the flow rate gets larger, the d_eff / L will get larger, but will never*

*get to 1 or higher, as a linear fit would imply. It seems that an exponential fit (d_eff / L = A – Be^(C\*flow_rate)) would be more appropriate.*

Thank you for the suggestion. The experimental data show that there is likely to be a linear relationship between d_eff/L and flow rate when the latter is in the range of 0.5-1.8 L/min, which is the possible range of the sampling flow for TD-CEAS. However, as you said, an exponential fit maybe be more appropriate if more experimental data, including very low and high flow rate, are available.

[Figure]

29. *Figure 7: The caption states that the orange columns correspond to HNO₃, but the legend indicates CH₃O₂NO₂. Which is correct?*

Thank you for pointing out the mistake. The orange columns correspond to $CH_3O_2NO_2$, and the name has been revised in Figure 7.

30. *Figure S5: How was this simulated? Was it checked experimentally? How do the authors reconcile this non-uniform temperature profile with their statement on line 152 that "it is assumed that the temperature of the heating part is uniform"?*

We measured the temperature profile of the cooling line after the heated tube (when the distance is greater than 35 cm) by insertion of the thermocouple when the flowing rate is the same value during the sampling, and the temperature profile of the heated tube is simulated according to previous reports (Sobanski et al., 2016; Thieser et al., 2016; Wild et al., 2014). The statement on line 152 that "it is assumed that "temperature of the heating part is ninform" is used to show the limit of the

residence time for the air flow. In order to avoid confusing readers, the statement is deleted.

31. *Figure 9: The y-axis label is confusing. Doesn't using the SPEC method mean that the resulting [NO₂] is simply [PN], without needing to subtract [NO₂]_ref?*
The y-axis label renames as 'PNs [ppbv]'. Here 'PNs [ppbv]' is determined by PNs channel and reference channel by the 'SPEC' method.

[Figure]

[Figure]

*Referee # 3*

*This paper describes a newly developed measurement system of NO₂, PNs and ANs in the atmosphere based on a thermal dissociation cavity enhanced absorption spectroscopy method (TD-CEAS). The authors evaluate characterization of this instrument and confirm the performance in field observations.*

*In an NO₂, PNs and ANs measurement system based on TD followed by NO₂ analyzer, NOx in the atmosphere interfere measured values of PNs and ANs. In this paper, in-depth evaluations for the interference were performed. As a result, TD-CEAS can measure ambient PNs and ANs concentrations using precise "correction factors". I recommend the manuscript to be published in AMT. However, I found several concerns to be published in the present form, so the authors should perform appropriate revisions sufficiently.*

Thanks for the positive comments. We responsed these comments carefully and revised our manuscript accordingly.

1.  *NO₂ detection: The use of a CEAS method would be novel. But advantages of the use of a CEAS are unclear, so the authors should state the advantages of a CEAS. What are the advantages of CEAS over LIF, CRDS and CAPS?*

    Thanks for the suggestion, CEAS is an absolute measurement technology to measure many trace gases, which has a good performance in NO₂ measurment with high precision and sensitivity, and comparable with the LIF, CRDS and CAPS.

CEAS measures the integral signals of light intensity in a certain spectral window with and without the target absorbors in the cavity, retives the concentration of target speices (such as $NO_2$) by spectral fitting, here we uses I_ambient as $I_0$ to first derive a corrected $\alpha(\lambda)$, and then uses this to calculate [PNs] and [ANs], which is able to reduce the uncertainty caused by differing $NO_2$ concentration. The comparison is shown in Sect. 4.2.

As summarized in Table 2, there are several typical technologies to measure organic nitrates based on the thermal dissociation method. TD-LIF is the pioneer to determine organic nitrates by measuring $NO_2$ produced through pyrolysis (Day et al., 2002), and the technology has been developed well and deployed in considerable campaigns (Di Carlo et al., 2013; Farmer et al., 2006). TD-LIF has a high time resolution and low detection limit, but the determination of $NO_2$ has to rely on extra calibration. TD-CIMS has a similar limitation as TD-LIF, and the method can measure some individual species of PNs, which need corresponding standards to be calibrated one by one (Slusher et al., 2004). CRDS, CAPS and CEAS are all cavity-enhanced techniques with high sensitivity and time resolution, of which CRDS and CAPS have been applied to detect $NO_2$ after ON pyrolysis. Specifically, in this study, the ONs and PNs are determined directly through broadband absorption measurement by CEAS, which can avoid the uncertainty caused by multiple spectral fitting and subsequent differential calculations. Overall, TD-CEAS has a detection capacity similar to that of TD-LIF and others. Recently, PERCA-CRDS was developed to indirectly determine PNs by measuring $NO_2$ through chemical amplification, which also showed high sensitivity, but the technology for atmospheric measurements needs to be studied further.

Table 2. Typical thermal dissociation methods to measure organic nitrates.

| Method | Targets | Time resolution | Detection limit | Accuracy | Reference |
|---|---|---|---|---|---|
| TD-LIF | ANs, PNs | 10 s | 90 pptv | 10-15% | Day et al., 2002 |
| TD-LIF | ANs, PNs | 1 s | 18.4, 28.1 pptv | 22%, 34% | Di Carlo et al., 2013 |
| TD-CIMS | PAN, PPN | 1 s | 7, 4 pptv | 20% | Slusher et al., 2004 |
| TD-CRDS | ANs, PNs | 1 s | 100 pptv | 6% | Paul et al., 2009 |
| TD-CAPS | PNs, ONs | 2 min | 7 pptv | N.A. | Sadanaga et al., 2016 |
| TD-CRDS | ANs, PNs | 1 s | 28 pptv | 6%+20 pptv | Thieser et al., 2016 |
| TD-CRDS | ANs, PNs | 1 s | 59, 94 pptv | 8%+10 pptv | Sobanski et al., 2016 |
| PERCA-CRDS | PNs, PAN | 1 s | 6.8, 2.6 pptv | 13% | Taha et al., 2018 |
| TD-CEAS | ANs, PNs | 6 s | 90 pptv | 9% | This work |

2. *Interference: The authors performed in-depth evaluations for the interference of NOx with PAN. But I could not find the evaluations for the interference with ANs. The authors should state the evaluations for the interference with ANs as well as PNs.*

We agree with this comment, it is very important to assess the interferences of NOx to ANs measurements. However the ANs source is not available to explore those interferences. Therefore, we did the model simulations and then established a look-up table by setting MeN as a representative of ANs.

*Other minor and technical comments:*

3. *lines 26-27, "alkyl nitrates (ANs, RONO$_2$)": There are many kinds of RONO$_2$ other than "alkyl" nitrates.*

Yes, but "alkyl nitrates (ANs, RONO$_2$)" is a collective term of RONO$_2$, and frequently used in previous reports about ANs measurement (Di Carlo et al., 2013; Keehan et al., 2020; Paul et al., 2009; Sobanski et al., 2016; Thieser et al., 2016), so here we follow these references.

4. *Line 64, "and cavity enhanced spectroscopy": Did the authors forget to delete?*

We revise "and cavity enhanced spectroscopy" as "Afterwards, chemical ionization mass spectrometry (CIMS), cavity ring-down spectroscopy (CRDS) and cavity attenuated phase-shift spectroscopy (CAPS) are used to quantify the pyrolysis products…"

5. *Line 175: The authors should define MCM. (Master Chemical Mechanism?)*

Corrected accordingly.

6. *First paragraph on page 10: The authors explain that the reason for the insufficient decomposition efficiency of PAN at 180 ℃ is due to the recombination of PAN. I think the effect of the PAN recombination can be reduced by increasing the pyrolysis time. What is the reason for not doing that (and making corrections)?*

That is a good question. The figure below is an example of model result to show the process of thermal dissociation of PAN in PNs channel. In fact, the PAN has been well dissociated in the heated channel in this instrument. Increasing the pyrolysis time will just increase wall loss of PA, but not very effective. Importantly, the recombination of PAN dominatingly happened after the heated channel. Therefore, the effective solution may be to (1) increase the wall loss of PA generated by pyrolysis in the heate tube and subsequent cooling tube, (2) to reduce the delivery time for gas to pass through the cooling tube.

The discussion is added in Sect. 5: "We highlight the impact of interference reactionreactions in heated channels for accurately measuring PNs and ANs. Although the look-up table can correct the interferences, the best way to reduce it them is to quench $RO_2$ in during the sampling process by improving the instrument design, such as by increasing the wall loss of $RO_2$ in the heated channel. "

[Figure]

Figure S3. A simulated example of the PAN pyrolysis in the PNs channel at 180 ℃ if the PAN source is equal to 4 ppbv. The concentration of relative species changes with the residence time, the red line is concentration of PAN, the blue line is the concentration of PA radical, and the yellow line is the concentration of $NO_2$. The red part in the plot is the duration time when the air flow goes through the heating part of quartz tube, and the blue part is the duration time when the air flow goes through the cooling part.

7. *Figure 7: Which is correct, $CH_3O_2NO_2$ in the legend or $HNO_3$ in the caption?*
   The orange columns correspond to $CH_3O_2NO_2$, and the name was corrected in Figure 7.

**References.**
Day, D. A., Wooldridge, P. J., Dillon, M. B., Thornton, J. A., and Cohen, R. C.: A thermal dissociation laser-induced fluorescence instrument for in situ detection of NO2, peroxy nitrates, alkyl nitrates, and HNO3, J. Geophys. Res.-Atmos., 107, 2002.

Di Carlo, P., Aruffo, E., Busilacchio, M., Giammaria, F., Dari-Salisburgo, C., Biancofiore, F., Visconti, G., Lee, J., Moller, S., Reeves, C. E., Bauguitte, S., Forster, G., Jones, R. L., and Ouyang, B.: Aircraft based four-channel thermal dissociation laser induced fluorescence instrument for simultaneous measurements of NO2, total peroxy nitrate, total alkyl nitrate, and HNO3, Atmos. Meas. Tech., 6, 971-980, 2013.

Farmer, D. K., Wooldridge, P. J., and Cohen, R. C.: Application of thermal-dissociation laser induced fluorescence (TD-LIF) to measurement of HNO3, Sigma alkyl nitrates, Sigma peroxy nitrates, and NO2 fluxes using eddy covariance, Atmos. Chem. Phys., 6, 3471-3486, 2006.

Friedrich, N., Tadic, I., Schuladen, J., Brooks, J., Darbyshire, E., Drewnick, F., Fischer, H., Lelieveld, J., and Crowley, J. N.: Measurement of NOx and NOy with a thermal dissociation cavity ring-down spectrometer (TD-CRDS): instrument characterisation and first deployment, Atmos. Meas. Tech.,

13, 5739-5761, 2020.

Fuchs, H., Dube, W. P., Lerner, B. M., Wagner, N. L., Williams, E. J., and Brown, S. S.: A Sensitive and Versatile Detector for Atmospheric NO2 and NOx Based on Blue Diode Laser Cavity Ring-Down Spectroscopy, Environ. Sci. Technol., 43, 7831-7836, 2009.

Jenkin, M. E., Saunders, S. M., and Pilling, M. J.: The tropospheric degradation of volatile organic compounds: A protocol for mechanism development, Atmos. Environ., 31, 81-104, 1997.

Keehan, N. I., Brownwood, B., Marsavin, A., Day, D. A., and Fry, J. L.: A thermal-dissociation-cavity ring-down spectrometer (TD-CRDS) for the detection of organic nitrates in gas and particle phases, Atmos. Meas. Tech., 13, 6255-6269, 2020.

Liu, J., Li, X., Yang, Y., Wang, H., Wu, Y., Lu, X., Chen, M., Hu, J., Fan, X., Zeng, L., and Zhang, Y.: An IBBCEAS system for atmospheric measurements of glyoxal and methylglyoxal in the presence of high NO2 concentrations, Atmos. Meas. Tech., 12, 4439-4453, 2019.

Osthoff, H. D., Brown, S. S., Ryerson, T. B., Fortin, T. J., Lerner, B. M., Williams, E. J., Pettersson, A., Baynard, T., Dube, W. P., Ciciora, S. J., and Ravishankara, A. R.: Measurement of atmospheric NO2 by pulsed cavity ring-down spectroscopy, J. Geophys. Res.-Atmos., 111, 2006.

Paul, D., Furgeson, A., and Osthoff, H. D.: Measurements of total peroxy and alkyl nitrate abundances in laboratory-generated gas samples by thermal dissociation cavity ring-down spectroscopy, Rev. Sci. Instrum., 80, 2009.

Saunders, S. M., Jenkin, M. E., Derwent, R. G., and Pilling, M. J.: Protocol for the development of the Master Chemical Mechanism, MCM v3 (Part A): tropospheric degradation of non-aromatic volatile organic compounds, Atmos. Chem. Phys., 3, 161-180, 2003.

Slusher, D. L., Huey, L. G., Tanner, D. J., Flocke, F. M., and Roberts, J. M.: A thermal dissociation-chemical ionization mass spectrometry (TD-CIMS) technique for the simultaneous measurement of peroxyacyl nitrates and dinitrogen pentoxide, J. Geophys. Res.-Atmos., 109, 2004.

Sobanski, N., Schuladen, J., Schuster, G., Lelieveld, J., and Crowley, J. N.: A five-channel cavity ring-down spectrometer for the detection of NO2, NO3, N2O5, total peroxy nitrates and total alkyl nitrates, Atmos. Meas. Tech., 9, 5103-5118, 2016.

Thieser, J., Schuster, G., Schuladen, J., Phillips, G. J., Reiffs, A., Parchatka, U., Pohler, D., Lelieveld, J., and Crowley, J. N.: A two-channel thermal dissociation cavity ring-down spectrometer for the detection of ambient NO2, RO2NO2 and RONO2, Atmos. Meas. Tech., 9, 553-576, 2016.

Wild, R. J., Edwards, P. M., Dube, W. P., Baumann, K., Edgerton, E. S., Quinn, P. K., Roberts, J. M., Rollins, A. W., Veres, P. R., Warneke, C., Williams, E. J., Yuan, B., and Brown, S. S.: A Measurement of Total Reactive Nitrogen, NOy, together with NO2, NO, and O-3 via Cavity Ring-down Spectroscopy, Environ. Sci. Technol., 48, 9609-9615, 2014.